# An Early Mid Latitude Aurora Observed by Rozier (Beausejour, 1780)

Chiara Bertolin[1], Fernando Domínguez-Castro[2,3], Lavinia. de Ferri[1]

[1]Department of Mechanical and Industrial Engineering, Norwegian University of Science and Technology, Trondeim, 7491, Norway
[2] ARAID Foundation, 50018, Zaragoza, SPAIN
[3] Departamento de Geografía y Ordenación del Territorio, Universidad de Zaragoza, 50009, Zaragoza, SPAIN

*Correspondence to*: Chiara Bertolin (chiara.bertolin@ntnu.no)

**Abstract.** Auroral observations are an uncommon phenomenon at low and mid latitudes that, at the end of the 18th century was not well known and understood. Low and mid geomagnetic latitude Auroral observations provide information about episodes of intense solar storms associated with flares and outstanding coronal mass ejection (CME) and on the variation of the geomagnetic field. However, for many observers at mid and low latitude, the features of a northern light were unknown, so they could easily report it as a phenomenon without explanation. In this work, we found that an earlier mid geomagnetic latitude aurora was observed in Beausejour, close to Béziers (43º 19′ N, 3º 13′ E), France, by the abbot Francois Rozier. He was a meticulous botanist, doctor and agronomist with special interest in atmospheric phenomena. On 15 August 1780, from 19:55 to 20:07 (Universal Time), Francois Rozier observed a "phosphoric cloud". A careful analysis of the report points out that he was reporting an auroral event. The recovery of auroral events at low and mid latitude during the 1780's is very useful to shed light to the solar activity during this period because there are few records of sunspot observations.

## 1 Background and Introduction

Incursions of high-energy particles from space, mainly solar wind, strongly interact with the Earth's magnetosphere, causing ionization and excitation of atmospheric gases and auroral emissions (Brekke, 2013; Gonzalez et al., 1994). The aurora borealis is a spectacular phenomenon that have been recorded from the Assyrians and Babylonians (Stephenson et al., 2004; Hayakawa et al., 2016; Hayakawa et al., 2019c) till nowadays. However, it is not until 1733 when Mairan (1733) suggested that the aurora could be caused by the solar atmosphere (Krivsky, 1984).

Low and mid latitude auroras (LMLAs) are usually associated with intense space weather events, frequently caused by coronal mass ejections (CME) (Gonzalez et al., 1994; Vázquez et al., 2006). This was the case of well studied extreme space weather events as those occurred on March 1582 (Hattori et al., 2019; Carrasco & Vaquero, 2020); September 1770 (Hayakawa et al., 2017a); the "Carrington event" August/September 1859 (Green and Boardsen, 2006; Green et al., 2006; Humble, 2006; Tsurutani et al., 2003; Cliver and Dietrich, 2013; Hayakawa et al., 2019a); the storm on 1872 February (Hayakawa et al., 2018; Silverman, 2008); the extreme event on September 1909 (Hayakawa et al., 2019b); May 1921 (Hayakawa et al., 2019a; Hapgood, 2019; Silverman and Cliver, 2001; Love et al., 2019) or March 1989 (Allen et al., 1989;

Boteler, 2019) resulting in extreme magnetic disturbances and auroral displays at very low latitudes. It is important to note that extreme space weather events of these magnitude can provoke important impacts on our highly technological dependent society, especially in activities related with the aviation, the GPS signals, radio communication, and the electric power grid

(Baker et al., 2008; Riley et al., 2018).

Low and mid latitude aurorae show an association with solar activity indices as sunspot records. This link has been observed during the telescopic era (Silverman, 1992; Lockwood and Barnard, 2015; Lockwood et al., 2016) but also in pre-telescopic era from the comparison with naked-eye sunspot reports (Hayakawa et al., 2017a; Bekli and Chadou, 2019). This relationship is due mainly to the highest frequency of LMLAs during the maximum and the decaying phase of the solar cycle

(Gonzalez et al., 1994). Therefore, the mid-latitude aurorae, being footprints of solar CMEs, can be considered as proxies for the long-term solar activity

Auroral catalogues are important tool to understand the long-term interaction among the solar activity and the Earth's magnetosphere (Legrand and Simon, 1987; Silverman, 1992). In the last centuries many auroral catalogues have been developed e.g. Mairan (1733), Lovering (1866), Fritz (1873), Angot (1897), Tromholt (1902), Link (1962, 1964), Krivsky &

Pejml (1988), Loysha et al. (1989) or Ordaz (2010). LMLAs, although rare, are recorded in these catalogues by professional and non-professional observers. During the 18th century, in Europe, there were some professional observers who were familiar with the phenomenon and who recorded auroras systematically e.g. Francisco Salvà (Barcelona, Spain) (Vaquero et al., 2010) or Giuseppe Toaldo (Padova, Italy) (Domínguez-Castro et al., 2016). Nevertheless, there were many sporadic observers that also recorded LMLAs unknowingly, cataloging them as strange and inexplicable phenomena. These sporadic

reports are important to generate and extend LMLAs catalogues but require an accurate analysis to avoid possible misinterpretations (Kawamura et al., 2016; Usoskin et al., 2017; Stephenson et al., 2019). Here, we analyzed an observation made by the Abbot Francois Rozier in 1780 with enough details and quantifiable information to understand if he observed a LMLA or a different phenomenon.

**2 Methodology**

55 1.

2.

### 2.1. The Observer

Jean-Baptiste François Rozier, (Lione, 23 January 1734 - Lione, 29 September 1793) (Fig. 1a) (Gutton and Bonnet, 1991) was a professor of Botanic and Medicine at the University of Lione. He studied at the Jesuit college at Villefranche-sur-

Saône and at theSaint-Irénée seminary in Lyon. In 1771 Rozier moved to Paris to edit the *Journal de Physique et d' Histoire Naturelle* founded by Jacques Gautier d' Agoty; after becoming the journal owner he renamed it as *Journal d' Observations sur la Physique, l' Histoire Naturelle et sur les Arts et Métier*s and later as the *Journal de Physique* where the original versions of many fundamental memoirs appeared (McKie, 1957). Rozier maintained the journal up to 1779 when he devoted himself to the writing of the *Cours d' agriculture* a periodical that was edited by his nephew, the mineralogist and priest

Jean-André Mongez (21 November 1750 – May 1788). In 1779 he became prior of the abbey at Nanteuil-le-Haudouin (located between Paris and Reims), while in July 1780 Rozier bought an estate close in Beauséjour, in the suburb of Béziers (43°19′ N, 3°13′ E), Southern France (domaine de Beauséjour) to install his own model farm (1781). Here he could edit his *Cours Complet d' Agriculture Théorique et Pratiqueou Dictionnaire Reisonné et Universel d' Agriculture* (twelve volumes in form of a dictionary, of which nine were by Rozier himself, 1781–1800, and the last two were published after his death).

Finally, he sold the property and in 1786 moved to Lyon where he accepted a position as Director of the School of Agriculture. Finally, he became constitutional curate of Sainte-Polycarpe parish in Lyon and was killed during the siege of the town the night between the 28th and 29th of September 1793 (French Revolution).

Rozier was a member of the Académie de Lyon and thanks to his activity as editor of scientific journals was in contact with the most famous scientists and intellectuals of his times. He devoted his life to the observation of botanical or agricultural

biological, chemical, physical and meteorological phenomena.

1.

2.

    **2.1.**

**2.2. The Documentary Source and the Observation description**

The observation was described in the "*Observations sur la physique, sur l'histoire naturelle et sur les arts, avec des planches en taille-douce*" tome XVIII under the title "*Observation sur une Nuée rendue phosphorique par une surabondance de l'électricité, vue de Beauséjour près de Béziers, le 15 Août* " [About a cloud rendered phosphoric by an overabundance of electricity observed at Beausejour the 15th of August] (Rozier, 1781) (Fig. 1). The most important fragments of the

observation are reported below in English, while the complete original French version is reported in Figure 1.

"*The closer the night approached, the more the clouds were pushed and accumulated towards the great chain of mountains of the third order that cross the low-Languedoc from east to west... At 20:05 it was completely night. It was at this moment that, examining the direction and the effects of the flashes, I noticed behind the slope of the hill, which on one side blocks the view from my house, a bright spot. This light did not look like that of a candle seen from afar, nor that which spreads from a*

*forest or grass when they are set on fire. It seemed to me to have the whitish color of phosphorus burning in the open air, or rather of that of mercury stirred in a tube without air. This bright spot gradually acquired volume and intensity. It imperceptibly formed an area, a phosphoric band that appeared to my eyes at a height of 3 feet: and starting from the top of the hill almost to Béziers, this area seemed to form the base of a 60° angle, whose summit responded to my eye.*

*On this first luminous area, a second one of the same height formed, and it had only 30° of extension [width], or half of that*

*of the lower area. Between them remained a void whose height equaled that one of the two areas considered separately.*

*Even if these two zones followed a horizontal direction, it is not to believed that their line of demarcation followed exactly a straight line. We noticed on both some irregularities, roughly as on the edges of that big white cloud, storm forerunners, and these edges were not all equally bright even if the center of the zones showed a uniform light.*

*During the period of time when these areas were moving eastward, the lightning and thunder noise were more rapid; finally,*

*at three different times, a flash started from the end of the lower area. But an object worthy of note is that the noise following these flashes, if there was one, was weak and I would dare to say almost null because I could not distinguish it from the noise of the thunder that was starting from the upper region and from a greater distance. Every flash, launched by the general mass, made me clearly appreciate the vines, the crops, the top and the sinuosity of the small mountains located in front of the big chain.*

*That light helped me to understand that the areas were closer to me and did not belong to the mass of clouds pushed by the winds towards the mountain.*

*This phenomenon was shown from 20.05 until 20.17. In this instant a blow of wind from the south changed the direction of the clouds, bringing them closer to the big mountain chain, and the storm moved away from Beziers.*

*It would seem that these areas were a simple mass of vapors, only charged by electricity, which made them transparent and*

*phosphoric. It is proven by the fact that three times the flash disappeared and the trail of light left by the flash appeared to be more than twice the diameter of normal flashes. The [apparent] proximity of the objects could, it is true, be due to these optical effects.*

*I am led to believe that these areas were detached entities [bodies] and that they did not belong to the mass of the other clouds because the mountains were visible behind them when the flash departed from the big mass; finally, when the flash*

*started from these areas, there was no explosion.*

*I don't know if such a phenomenon has been observed elsewhere; but I never read anything that can be compared to it."*

**3 Analysis of the Observation**

Hour of observation and sun depression angle: Rozier describes the starting (20:05) and ending (20:17) hour of his observation as local solar time (LST) i.e. the measure of local time as in use in the XVIII century. The pendulum clocks

locally could be synchronized following the daily data reported in the Ephemerides with the time of the sunrise, midday and sunset published yearly (Jeaurat, 1780). Given its longitude, these times correspond to 19:55 and 20:07 in Universal Time (UT) respectively. At these times the solar depression angle was 11.6° and 13.4° respectively. Therefore, although Rozier described the observational conditions as if it was "completely night", however the observation started during the nautical twilight and concluded in the astronomical twilight. As reported in Silverman and Cliver, 2001; Vaquero et al. 2008;

Hayakawa et al. 2019, some bright aurorae were seen under twilight. The calculation of the solar depression angle for the

geographical coordinates in Beausejour and the day of the observation has been performed using the HORIZONS Web-interface of the American National Aeronautics and Space Administration (NASA) (https://ssd.jpl.nasa.gov/horizons.cgi?s_type=1#top).

Shape: Related to the shape description, Rozier was very accurate. The main structure described by Rozier is: "*it formed a zone, a phosphoric band…at a height of 3 feet….and finally it formed an angle of 60°…. above this first luminous zone a second* [zone] *of the same height was formed, but with 30° of extension only i.e. half of that of the lower zone. Between one and the other a void remained, the height of which matched that one of the two connected zones*". This description may fit with the report of the auroral forms class without ray structure i.e. homogeneous arcs or uniform diffuse surface, and

homogenous bands too (Störmer, 1955). Nevertheless the beginning of the aurora it could resemble some aspect of an auroral sub-storm expansion (Ebihara et al., 2017; Stephenson et al., 2019) which is characterized by initial brightening of aurora, followed by a bulge expanding in all directions (Akasofu, 1964; Akasofu et al., 1965). In fact, Rozier reports: "*I noticed a luminous point…this luminous point acquired slowly* [over time] *volume and intensity*". Furthermore, he records some flaming during the event "*in one or in the other zone I noticed irregularities, as well as on the edges of those big white*

*clouds* [i.e. general mass or bulge]. He finally adds more details: "*This edge was not homogeneously bright, although in the center presented uniform bright. In the time over which the zones moved Eastward…. a flash started from the end of the lower area* [of the general mass or bulge]".

Color: He carefully mentioned the color: "*whitish color of phosphorus burning in the open air*". As stated by Stephenson et

al. (2019) at Low-Mid geomagnetic latitude, northern lights have generally higher probability to be observed when they are reddish. While, when they appear whitish to the human eye, the aurorae have lower brightness. In addition, such effect of the human eye is enhanced if the moon is also present in the sky as the eye cannot be "dark adapted". Moreover, the whitish auroral color may be explained with the enhancement of the 557.7 nm of Oxygen with weak brightness or due to the Oxygen mixture with other emissions as well (Ebihara et al., 2017; Stephenson et al., 2019). Examples of observation which confirm

that LMLA are whitish in color during extreme space weather events are reported by Ebihara et al. (2017), Green and Boardsen (2006), Hayakawa et al. (2017b) or Willis et al. (1996). Rozier observed a white aurora, this made the phenomena more unusual and increase the possibility of misinterpretation of the phenomenon by Rozier himself.

Noise: Silverman and Tuan (1973) said that from observational evidence, the most likely sound accompany auroral observations could derive from discharges generated by aurorally associated electric fields. Rozier, although in his

observation reported: "*It appeared* [to me] *that these areas were a simple mass of steam, only charged by electricity, which it made them transparent and phosphoric*". However, he concluded saying that: "*for three different times, a flash, with almost null noise, started from the end of the lower area* [i.e. the bulge] … [and again] *when the light flashed …there was no*

*explosion"*. This absence of sound recorded by Rozier discard a possible misinterpretation with other noisy atmospheric phenomena.

Moon: whether or not an aurora is overshined by the moon depends on the lunar phase, the brightness of the aurora, and the angular distance between the moon and the sky position occupied by auroral emission (Stephenson et al., 2019). Rozier does not report any information about the moon. But the moon was in the sky that day. The moon, on 15 August 1780 was full moon and rose at 19:25 (UT) at an azimuth angle of 111.4° ESE direction, i.e. opposite respect the direction of observation of Rozier and close to the horizon. During the time Rozier observed the phenomenon, the moon was at azimuth angle 116.5°

and elevation angle 3.4° (at 19:55 UT) while at the end of his observation it was at azimuth angle 118.5° and elevation angle 5.3° (at 20:07 UT) therefore always in the direction ESE. The short time of the observation suggests that although the aurora was highly bright because Rozier could record it with full moon in the sky (Stephenson et al., 2019; Hattori et al., 2019). However, because of the moon rise above the horizon, the light conditions could have fainted the auroral display and made it apparently whitish due to their relative brightness. In literature several auroral observations are reported during full moon

e.g. those observed on the 18 February and on 12 November 1837 (Olmsted, 1837; Snow, 1842), those reported by Martin (1847) and Glaisher (1847) on October 24 of that year, and the event observed on 4 September 1908 described by Barnard (1910).

Geomagnetic latitude: We have calculated the temporal evolution of the geomagnetic latitude in Beziers for the night of the

observation using the geomagnetic model gufm1 (Jakson et al., 2000). The geomagnetic latitude, $\varphi$ equals to 50,18°N is obtained by equation (2) in the hypothesis of a dipolar configuration for the geomagnetic field.

$$\varphi = \frac{\tan I}{2} \qquad (2)$$

where I is the magnetic inclination obtained from the gufm1 model for the year 1780. This implies that the aurora is in the lower limit of the mid-latitude aurora or at the border for being defined LLA as this threshold often in literature is around

50,00°N of geomagnetic latitude.

**4 Discussion**

In the previous section we have verified that Rozier observed an aurora the night of the 15 August 1780. According to the Angot catalogue (Angot, 1897) in this night the aurora was also observed at Ratisbon (Germany, 49° 01' N, 12° 05' E), 5.5°

further north than Beziers. The Angot catalogue has been extensively used on the reconstruction of auroral nights and as a proxy of the long-term geomagnetic variability. Nevertheless, it is important to note that Angot (1897) is a secondary source (the author was no witness of the facts he describes) and does not provide information on the primary sources he consulted for the elaboration of the catalogue. Secondary sources must be used carefully because can include errors due to the transcription or interpretation of the primary source. For this reason, it is valuable to found primary sources that corroborate

the information provided by Angot, specially during the nights in which Angot recorded an event in a single location, as the night of 15 August 1780.

In addition, magnetic indexes are not available in 1780. The Ak index is in fact available since 1844 (Nevanlinna & Kataja, 1993), the aa index since 1868 (Mayaud, 1980) and the geomagnetic IDV index (Svalgaard & Cliver, 2010) is available since 1835. For this reason, LMLAs catalogues and sunspot number are used here as proxies of the geomagnetic activity at

Rozier's times.

First the aurorae catalogue at comparable latitudes have been analyzed. Then, they were compared with two existing coeval series of auroras homogeneously recorded at low latitude by trained observers as Toaldo (1766-1797) (Padova, Italy 45º 24' N, 11º 52' E) (Domínguez-Castro et al., 2016) and Salvà (1780-1825) (Barcelona, Spain 41º 23' N, 2º 10' E) (Vaquero et al, 2010). For differentiation purpose, the additional series of auroras observed by Thomas Hughes from Stroud, United

Kingdom (mid-latitude 51,75°N, 2,22°W) (Giles, 2005) has been also reported. Figure 2 shows the Toaldo, Salva and Hughes yearly total observations of auroras and the geomagnetic latitude respectively in Padua, Barcelona, Stroud and Beziers over the common 1766-1800 period. The Rozier's observation was close to the maximum LLA observed by Toaldo in Padova (1779). Nevertheless, no aurora was recorded by Salvà at Barcelona during 1780. At higher latitudes (Stroud) Hughes recorded a mean low activity of auroras for that year.

Figure 3 shows the sunspot number during the period 1766-1800. Rozier´s observation was in the declining phase of the solar cycle 3, 2-years after the maximum. This is a good moment to see LMAA because long-lived coronal holes - source of high ionized particles in the solar wind - occur more frequently in the declining phase of the sunspot cycle (Verbanac et al., 2011; Lefèvre et al., 2016). It is important to note that the Rozier's observation occurred in a period with few sunspot records. As we can see in Figure 4 the solar observations during the 1780`s are rare, frequently below the 30 observations

per year.

Figure 4 shows at monthly resolution the solar activity and the auroras recorded in Padova from August 1779 to August 1781. We can see that no aurora was observed during August and only one solar observation was recorded in this month. The nearest solar observation was the 30th of August when J.C. Staudach report 4 groups in the solar disk. The previous observation was done by P. Zeno at 12th of July recording one group (Vaquero et al., 2016). It means this event occurred in

an interval without sunspots data for 48 days. The scarcity of solar observations makes this study extremely important. Because the understanding of the space climate in this period improves the discussions on the transition from the high solar cycles to the Dalton Minimum (Usoskin et al., 2009; Karoff et al.,2015; Owens et al., 2015; Hayakawa et al., 2020).

**Conclusions**

We have found a record of an atmospheric phenomenon observed on 15 August 1780 in Beausejour, close to Béziers (43º 19′

N, 3º 13′ E), France, by the abbot Francois Rozier described as a "*big white cloud … whitish color of phosphorus burning in the open air*". Rozier was not an astronomer and it is clear that he did not fully understand the phenomenon he was

recording. Probably for this reason he recorded the event with minute details to later discuss it with other academicians of his time. Thanks to this accuracy, we have been able to analyse quantitative information and facts that contribute to confirm that Francois Rozier observed a Mid latitude aurora that night. The aurora was observed during the nautical and astronomical twilight, it was white, enough brilliant to not be overshined by the full moon which however was above the horizon in ESE direction. It showed two bands and some rays which could fit with the class of auroral forms of both homogeneous arcs/uniform diffuse surface, and homogenous bands. Its temporal evolution could also resemble an auroral sub-storm expansion.

This auroral event contributes to enlarge the geomagnetic knowledge of the late 18th century period in which the geomagnetic and the solar activity have high uncertainties due to few sunspots and LMLA observations reported from primary sources.

The Rozier record is a clear case of how, a scientist from a research field far from astronomy or meteorology in the 18th century, could record and publish descriptions on atmospheric phenomena that he did not fully understand but however he considered worth to be documented. These sources are very valuable because they report details of infrequent and/or partially unknown atmospheric phenomena. In this case the Rozier's report had contributes to enlarge the geomagnetic knowledge of a period with low information.

**Data availability**

The datasets generated for this study are available on request to the corresponding author.

**Authors contributions**

C. Bertolin conceived the study, performed the analysis and drafted the manuscript with F. Dominguez-Castro, who wrote the final manuscript. L. de Ferri translated the original data and conducted the historical research used in the study as well as contributed to scientific discussion of the article together with C. Bertolin and F. Dominguez-Castro.

**Competing Interests**

The authors declare that the research was conducted in the absence of any commercial or financial relationships that could be construed as a potential conflict of interest.

**Acknowledgment**

Credits for the use of Sunspot data to the World Data Center SILSO, Royal Observatory of Belgium, Brussels (http://www.sidc.be/silso/).

The research reported in this publication was supported through the financial support guaranteed by the Onsager Fellowship – Research Excellence Program at the Norwegian University of Science and Technology (NTNU) in Trondheim and the DRO500 project (PID2019-108589RA-I00) financed by the Ministry of Science and Innovation of the Spanish Government.

**ORCID iD**

Chiara Bertolin https://orcid.org/0000-0002-0684-8980

Fernando Dominguez-Castro https://orcid.org/0000-0003-3085-7040

Lavinia de Ferri https://orcid.org/0000-0001-8904-4779

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

# OBSERVATION

*Sur une Nuée rendue phosphorique par une surabondance de l'électricité, vue de Beauséjour près de Beziers, le 15 Août;*

*Par M. l'Abbé ROZIER, l'un des Auteurs de ce Recueil.*

LA chaleur de ce jour avoit été de 25 degrés & demi; le baromètre annonçoit un orage: de gros nuages blancs errèrent dans la région supérieure de l'atmosphère pendant toute la journée, & le soleil se coucha derrière une masse de ces nuages entassés les uns sur les autres.

A sept heures du soir, l'atmosphère se chargea de plus en plus; les nuages parurent quitter la région supérieure pour s'abaisser vers la terre, & un temps lourd & pesant étoit les moyens de respirer avec facilité. Plus la nuit approchoit, plus les nuages se ballonnoient & s'accumuloient vers la grande chaîne de ces montagnes du troisième ordre, qui traversent le Bas-Languedoc de l'est à l'ouest.

A sept heures trois quarts, la lumière incertaine de quelques éclairs partis du côté de l'ouest, annonçoit que le tonnerre grondoit à une distance trop éloignée pour être entendu. Peu-à-peu les éclairs furent multipliés, se succédèrent avec une rapidité surprenante, & le bruit du tonnerre commença à être sensible.

A huit heures du soir, les vents se contrarièrent & donnèrent aux nuages différentes directions; cependant, le vent d'ouest fut toujours le dominant. A cette époque, les coups de tonnerre redoublèrent du côté de la montagne, & le ciel étoit tout en feu. La nuit survint; il ne fut plus possible de distinguer la direction des nuages, parce que la vivacité de la lumière des éclairs faisoit paroître plus profonde l'obscurité qui lui succédoit; enfin l'orage s'élançoit avec rapidité de l'ouest à l'est, & il étoit terrible vers la montagne.

A huit heures cinq minutes, il étoit complètement nuit. C'est à cet instant, qu'examinant la direction & les effets des éclairs, j'aperçus derrière le penchant de la colline, qui, d'un côté, termine la vue de ma maison, un point lumineux. Sa lumière ne ressembloit pas à celle d'une bougie vue de loin, ni à celle que répandent du bois ou des herbes dans leur état d'ignition. Elle me parut avoir la couleur blanchâtre de celle du phosphore qui brûle à l'air libre, ou plutôt de celle du mercure qu'on agite dans un tube privé d'air.

Ce point lumineux acquit peu-à-peu du volume & de l'étendue. Il

forma insensiblement une zone, une bande phosphorique qui se montroit à mes yeux sur une hauteur de trois pieds: & en partant de la croupe de la colline pour s'approcher près de Beziers, cette zone sembloit former la base d'un angle de 60 degrés, dont le sommet répondoit à mon œil.

Sur cette première zone lumineuse, il s'en forma une seconde de la même hauteur, & qui n'avoit que 30 degrés d'étendue, c'est-à-dire, la moitié de celle de la zone inférieure. Entre deux, resta un vuide dont la hauteur égaloit celle d'une des deux zones prise séparément.

Quoique ces deux zones suivissent une direction horizontale, il ne faut pas croire que leur ligne de démarcation fût exactement en ligne droite. On remarquoit sur l'une comme sur l'autre des irrégularités à-peu-près comme sur les bords de ces gros nuages blancs, avant-coureurs de l'orage, & ces bords n'étoient pas tous également lumineux, quoique le centre des zones offrît une clarté uniforme.

Pendant le temps que ces zones avançoient vers l'est, les éclairs & le bruit des tonnerres se succédoient avec la plus grande rapidité; enfin, à trois reprises différentes, la foudre s'élança de l'extrémité de la zone inférieure. Mais un objet digne de remarque, est que le bruit qui suivit ces éclairs, s'il y en eut un, fut foible, & j'ose dire presque nul, parce qu'il ne me fut pas même possible de le distinguer du bruit des coups de tonnerre qui partoient de la région supérieure & dans un plus grand éloignement. Chaque éclair, lancé de la masse générale, me faisoit apercevoir très-clairement les vignes, les moissons, la croupe & les sinuosités des petites montagnes placées sur le devant de la grande chaîne. Cette lumière me servir à déterminer que les zones étoient plus rapprochées de moi, & ne faisoient pas corps avec la masse des nuages ballottés par des vents auprès des montagnes.

Ce phénomène brilla depuis huit heures cinq minutes jusqu'à huit heures dix-sept minutes. A cet instant un coup de vent du sud fit changer la direction des nuages, les porta plus près de la grande chaine des montagnes, & l'orage s'éloigna de Beziers.

Il y a toute apparence que ces zones étoient un simple amas de vapeurs tellement chargées de l'électricité, qu'elles les rendoient transparentes & phosphoriques. Ce qui le prouve, c'est que trois fois la foudre en est partie, & la trainée lumineuse qui formoit l'éclair a paru d'un diamètre plus que double de celui des éclairs ordinaires. Le rapprochement des objets pouvoir, il est vrai, avoir part dans cet objet d'optique.

Je suis fondé à croire que ces zones étoient des corps détachés & ne tenoient pas à la masse des autres nuages, puisqu'on distinguoit parderrière elles les montagnes lorsque les éclairs s'élançoient de la grande masse; enfin, lorsque la foudre partit de ces zones, il n'y eut point d'explosion.

J'ignore si un semblable phénomène a été observé ailleurs; mais je n'ai rien lu qui puisse lui être comparé.

**Figure 1: The two printed pages reporting the aurora observation made by Abbot Francois Rozier, on 15 August 1780 in Beziers, France (Rozier, 1781). Source: Google Books  https://books.google.com.bn/books?id=-t3F48h7xKUC&printsec=frontcover&hl=it&source=gbs_ge_summary_r&cad=0**

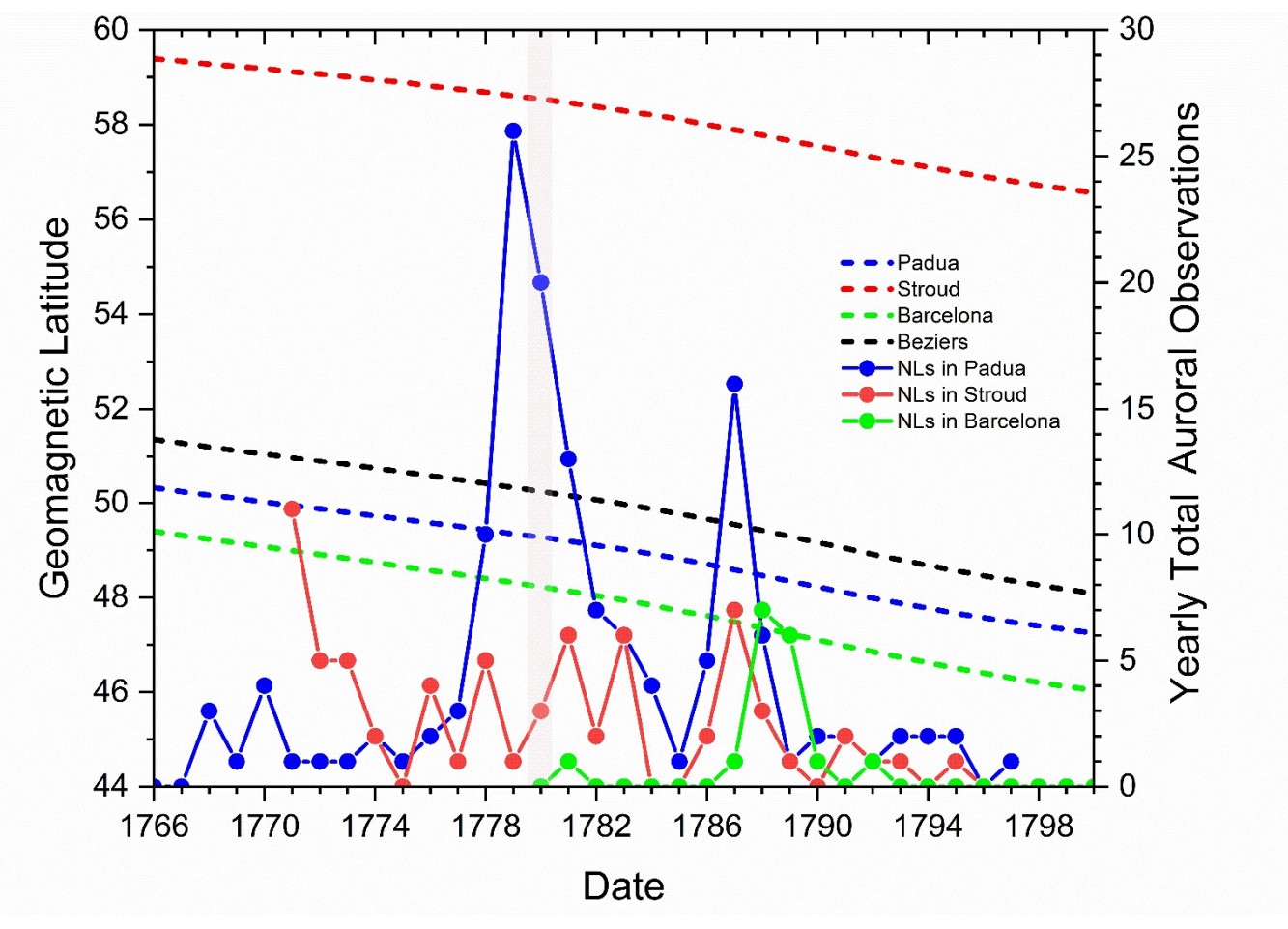


**Figure 2: Geomagnetic Latitude variations for Padua, Barcelona Stroud and Beziers and yearly total auroras recorded in these places by Toaldo, Salva and Hughes. The grey column indicates the year of the Rozier's auroral observation: 1780.**

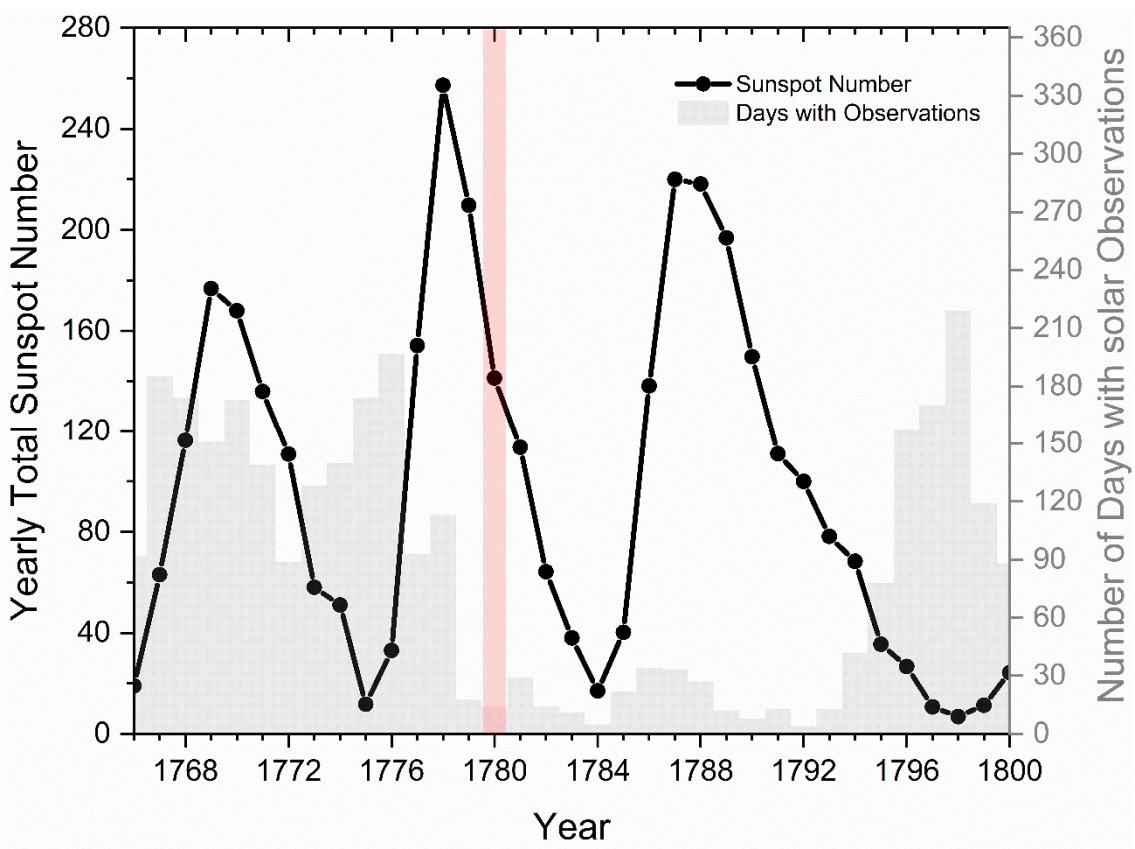

**Figure 3: Annual sunspot numbers and number of days with solar observations (SILSO –WDC**
       **; Clette et al.,2014; Clette and Lefevre 2016).**




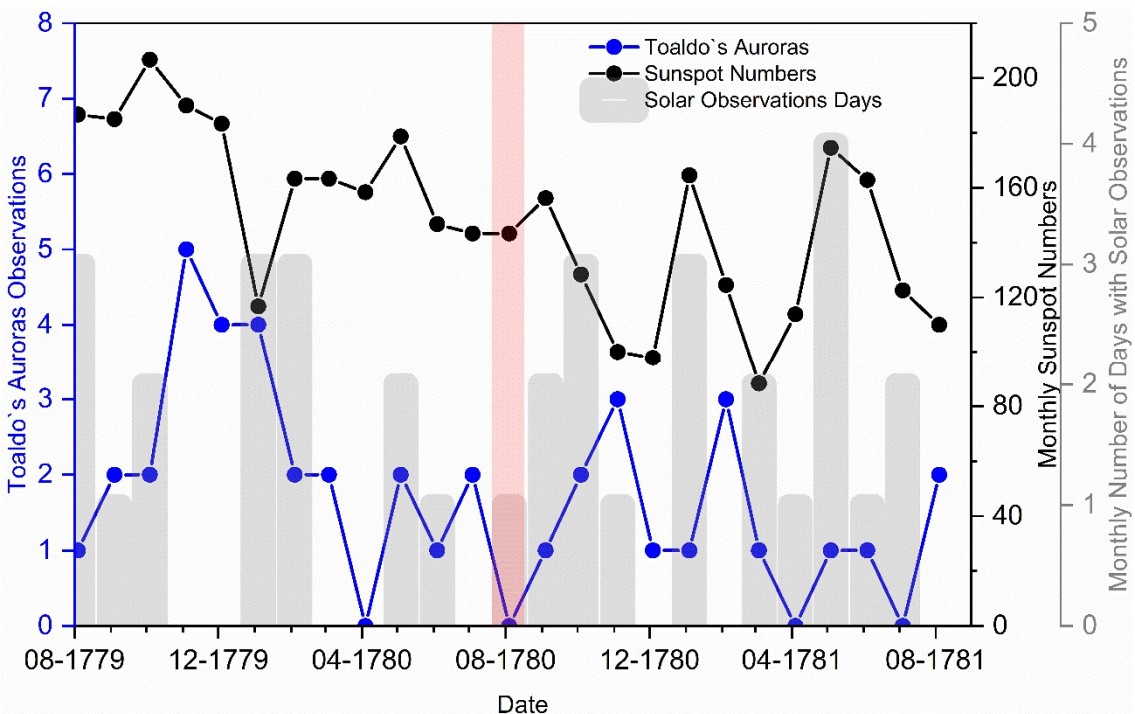

**Figure 4: Monthly sunspot numbers (Clette et al., 2014; Clette and Lefevre, 2016, http://www.sidc.be/silso/), days with solar observations and auroras from Toaldo catalogue from August 1779 to August 1781 (Dominguez-Castro et al., 2016; Vaquero et al. 2016).**