# Peer review of "An Early Mid Latitude Aurora Observed by Rozier (Beausejour, 1780)"

_Annales Geophysicae, 2020_

## Referee Comment (RC1) · Anonymous Referee #1 · 2 Mar 2020

I have carefully read the paper titled "An Early Low Latitude Aurora Observed by Rozier (Beziers, 1780)". The authors present a suspected aurora observed by Francois Rozier on 15 August 1780 in Beauséjour, close to Beziers (at MLAT= 50.18°N, according to the authors). It should be noted that the observation was made under adverse weather conditions (presence of a lightning storm). In section 4, the authors indicates that at the same time an aurora was also observed at Ratisbon (Germany, 49° N), 5.5° further north than Beziers, and recorded in Angot's catalogue (Angot, 1897). If this article is selected for publication, I suggest some revisions to the manuscript and other small suggestions before it can be published in ANGEO.

1 Background and Introduction Line 20: For the physical mechanism of the aurora origin, (Vazquez et al. 2014) is not the appropriate reference (see e.g., Brekke, A.,

2013, Physics of the Upper Polar Atmosphere, 2nd edn. Springer, Heidelberg). Lines 25-26: The three articles cited relate to the Carrington event. It is interesting to point out other exceptional events, such as that of 1921 (Silverman, S.M., Cliver, E.W.: 2001, J. Atmos. Solar-Terr. Phys. 63, 523), as well as that which occurred in 1770 (Hayakawa, H., et al.: 2017, Astrophys. J. Lett. 850, L31). Line 26: About LLA, the authors state that "and have been considered a proxy of solar activity". This needs to be correctly documented. Overall, I think that this section needs to be improved and expanded with more background information.

2 Methodology 2.1. The Observer The subsection (2.1), which is a biography of F. Rozier, is unnecessary and the text should be reduced considerably. Lines 35-39: please refer to reliable sources for accurate information and remove the links. 2.2. The Documentary Source and the Observation description Lines 65-66: The book's title should be rectified as follow: Observations sur la physique, sur l'histoire naturelle et sur les arts, avec des planches en taille-douce Lines 66-67: The subtitle should be rectified as follow: Observation sur une Nuée rendue phosphorique par une surabondance de l'électricité, vue de Beauséjour près de Beziers, le 15 Août Lines 82-83: bad translation: The sentence "avant-coureurs de l'orage" means "before the storm" not "before it was orange colored" Line 93: The sentence "l'orage s'éloigna de Beziers" means "the storm moved away from Beziers", not "the orange moved away from Beziers" Line 101: why the author uses the term "explosion"? Page 2: Footnote 1: The reference must be written correctly as indicated in "Manuscript preparation guidelines for authors" of ANGEO (Publisher, Location. . .). Also, please indicate the relevant pages. 3 Analysis of the Observation Line 105: Please specify how you obtained the two values of solar depression angle (13° and 14.9°). Lines 113-121: Color: as I said before, the orange color is not specified by the author. Therefore, this paragraph must be corrected. Lines 133-134: There is no exact definition of the low latitude, but for me the present event must be classified as a mid-latitude aurora!

4 Discussion Lines 138 and 140: (Angot, 1897) not (Angot, 1896) Lines 157-

160: A similar phase opposition and anti-correlation between auroral occurrence and sunspot were reported by some authors. It is an important point which must be well-documented (see e.g., Silverman, S.M., 1992, Secular variation of the aurora for the past 500 Years, Rev. Geophys. 30, 333–351).

Overall, a more extended state of the art is needed. Some articles relating to the present work should be viewed and cited (e.g., Ordaz, J., 2010, Auroras boreales observadas en la Península Ibérica, Baleares y Canarias durante el siglo XVIII, Treb. Mus. Geol. Barcelona 17, 45-110; Legrand, J. P., & Simon, P. A., 1987, Two hundred years of auroral activity (1780-1979), AnGeo 5, 161-167; …)

Conclusions I think the conclusion is too short and it does not summarize the work in sufficient detail.

References Line 237: The source of the data (WDC-SILSO) must be cited properly as indicated on their website. I think: SILSO data/image, Royal Observatory of Belgium, Brussels. In addition, you can also indicate the version.

Figure 1 is not cited in the text. Furthermore, Figure 1 (b) hides part of Figure 1 (a); I think it is better to remove Figure 1 (b).

---

## Referee Comment (RC2) · Anonymous Referee #2 · 4 Mar 2020

**Referee Report on MS angeo-2020-1 "An Early Low Latitude Aurora Observed by Rozier (Beziers, 1780)" by Bertolin *et al.**

**General Comments**

This is an interesting case report for the mid-latitude aurora on 1780 August 15 likely at Beauséjour near Béziers. The mid-latitude auroral report in 1780 is especially important, as this will be a footprint of solar eruption and a hint for contemporary solar activity where we do not have enough coverage of sunspot observations. Overall, I think this manuscript will be an interesting contribution, while further clarifications and explanations are needed both on its background and discussions. I have listed my comments below and wish the authors to address them. Their language needs to be improved as well, preferably with professional grammatical proofreading.

**Specific Comments**

**1. Introduction**

The scientific background of this article should be improved. Rather than associating "Incursions of high-energy particles from space, mainly solar wind, strongly interact with the Earth's magnetosphere" with the cause of auroral display, I would explicitly mention the coronal mass ejection with southward interplanetary magnetic field as a cause of auroral displays in low to mid magnetic latitude. I advice the authors to cite Gonzalez *et al.* (1994) and Daglis *et al.* (1999) for its references, rather than Vazquez *et al.* (2014).

The space weather hazards are not only geomagnetically induced currents, but also satellite drags (Oliveira and Zesta, 2019) or atmospheric radiations (Dyer *et al.*, 2018). These details should be cited with actual cases reports/predictions (Nakamura *et al.*, 2018; Love *et al.*, 2018; Boteler, 2019) and reviews (Pulkkinen *et al.*, 2017; Ngwira and Pulkkinen, 2018; Riley *et al.*, 2018; Oliveira *et al.*, 2018).

Given the magnetic latitude of Béziers (50.2° MLAT), I would consider this aurora not as "low-latitude aurora" but as "mid-latitude aurora". In comparison with auroral ovals during the extreme storms like the Carrington event (~30° MLAT), I consider that this extent is confined in mid magnetic latitude.

The references on the Carrington event should be updated. The reference of Green et al. (2014) is probably Green et al. (2006), as long as checking the NASA ADS database. Three benchmark articles for this event should be cited in this context (Tsurutani et al., 2003; Cliver and Dietrich, 2013; Hayakawa *et al*., 2019a). Moreover, recent studies have located at least three rivaling storms with extremely low-latitude auroral visibility: 1872 Feb (Silverman, 2008; Hayakawa *et al*., 2018) and 1921 May (Silverman and Cliver, 2001; Hapgood, 2019; Love *et al*., 2019). These cases should be documented as well.

The usage of LLA as historical solar activity is another story. I would rephrase this as "Being footprints of solar eruptions, the mid-latitude aurorae (or low-latitude aurorae) are considered as proxies for the long-term solar variability", citing several relevant articles such as Silverman (1992), Lockwood and Barnard (2015), Lockwood et al. (2016), Vázquez et al. (2016), and Hayakawa et al. (2017). Caveats must be noted here, however. Even when the solar activity is low, several great magnetic storms with significant auroral displays are reported as well (Garcia and Dryer, 1987; Hayakawa *et al*., 2020). This caveat should be clarified too.

**2. Methodology**

It is nice to cite Rozier's portrait and personal detail in this article. However, citing them from wikisource or other online resources is not the best scientific practice. Please specify their original references in the publications and cite them accordingly. The reference clarifications are especially important as most of P2 of this manuscript is devoted to its explanation and the readers may wish to know more about him with appropriate references.

**3. Analyses of the Observations**

The analyses seem sound but some improvements seem advised. The description of ""a flash started from the end of the lower area…". This is a frequent structure of the aurorae (Vaquero & Vasquez, 2009)" may be flaming of auroral display (Störmer, 1955). If the description of "a main structure of two bands oriented east to west" means westward auroral motion, this sounds consistent to the westward traveling surge

(Ebihara and Tanaka, 2015).

The whitish auroral colour are explained with "the enhancement of the 630,0 nm [OI] emission caused by soft electrons (<100 eV) precipitating from the plasmasphere" in this manuscript. However, I suspect the whitish colour may be explained to the enhancement of the 557.7 nm of Oxygen with weak brightness or its mixture with other emissions as well (*e.g.*, Ebihara *et al*., 2017; Stephenson *et al*., 2019). Rather than citing Abott and Juhl's statistics, it would be more straightforward to cite actual observational cases of whitish aurorae (See Section 6 of Stephenson *et al*., 2019).

For the sunspot number analyses, the authors need to cite the data source "WDC SILSO" appropriately. I would strongly recommend the authors to cite Clette et al. (2014) and Clette and Lefevre, 2016) for this dataset. Likewise, the authirs need to mention the WDC SILSO in the acknowledgment. I suspect that the cause of this storm is probably better explained with the coronal mass ejections (see Gonzalez *et al*., 1994; Daglis *et al*., 1999) rather than the high-speed solar wind from the corona hole.

**Conclusion**

The conclusion needs to be more developped to be an independent original article.

**Supplement**

It may be helpful to reproduce the French text here, as the authors stated "complete original French version is reported in the Supplementary Materials".

**Minor Comments**

P1L12: The authors need to be consistent for the usage of Béziers or Bezier. By the ways, isn't the observational site Beauséjour? If this is the case, the coordinate should be N43°19′, E3°13′.

P1L20: disturbs => disturbances

P1L22: Babylonians (Stephenson *et al*., 2004) => Assyrians and Babylonians (Stephenson *et al*., 2004; Hayakawa *et al*., 2016, 2019b)

P1L30: "require an accurate analysis to avoid possible misinterpretations" => Cite Kawamura *et al*. (2016), Usoskin *et al*. (2017), and Stephenson *et al*. (2019) here.

P2: Please italicise the journal titles.

P2L48: Béziers (Beauséjour) => Beauséjour in the suburb of Béziers

P3L68: "reported below in English" => "reported below in our English translation"

P3L95: "proved" => "proven"

P3L104: "the measure of time of the French Hours that lasted until the French Revolution in 1789" => Please cite a reference for this statement.

P3L104: "These times correspond" => "Given its longitude, these time stamps correspond"

P3L105: "nautical twilight" => "astronomical twilight"

P4L122-123: For auroral audibility, cite the review of Silverman and Tuan (1973).

P5L124-128: For bright aurorae visible during night with the full moon, it would be advised to reinforce the existing discussions with actual observational cases cited in Stephenson *et al*. (2019) and Hattori *et al*. (2019).

P5L137: "15th august 1780" => "15 August 1780".

P5L143: "is" => "it is"

P6L157: "decrease phase" => "declining phase"

P6L158: "2-years delayed respect the peak of the highest solar activity" => "2-years after the maximum"

P6L160: Cite Lefevre *et al*. (2016) here.

P6L160: "few solar observation" => "few sunspot observations"

P6L162: "very welcomed" => "useful" or "informative"

P6L164: "this solar observation" => maybe "the nearest solar observation"?

P6L166: "This is 48 days without sunspot information" => "It means this event occurred in an interval without sunspot data for 48 days".

Figure 1: The two figures are overlapped. They should be separated at least. The data source of the photograph should be addressed not with the URL but with shelf mark in the Library of Congress Prints and Photographs Division Washington.

Figure 3: Cite Clette *et al*. (2014) and Clette and Lefevre (2016) here.

Figure 4: Cite Dominguez-Castro *et al*. (2016) and Vaquero *et al*. (2016) here.

**References**

Boteler, D. H. (2019) A 21st Century View of the March 1989 Magnetic Storm, *Space Weather*, **17**, 1427-1441. DOI: 10.1029/2019SW002278

Clette, F., *et al*.: 2014, Revisiting the Sunspot Number. A 400-Year Perspective on the Solar Cycle, *Space Science Reviews*, **186**, 35-103. DOI: 10.1007/s11214-014-0074-2

Clette, F., Lefèvre, L.: 2016, The New Sunspot Number: Assembling All Corrections, *Solar Physics*, **291**, 2629-2651. DOI: 10.1007/s11207-016-1014-y

Cliver, E. W., Dietrich, W. F. (2013) The 1859 space weather event revisited: limits of extreme activity, *Journal of Space Weather and Space Climate*, **3**, A31. DOI: 10.1051/swsc/2013053

Daglis, I. A. (1999) The terrestrial ring current: Origin, formation, and decay, *Reviews of Geophysics*, **37**, 407-438. DOI: 10.1029/1999RG900009

Dyer, C., *et al*. (2018) Extreme Atmospheric Radiation Environments and Single Event Effects, *IEEE Transactions on Nuclear Science*, **65**, 432-438. DOI: 10.1109/TNS.2017.2761258

Ebihara, Y., Tanaka, T. (2015) Substorm simulation: Formation of westward traveling surge, *Journal of Geophysical Research: Space Physics*, **120**, 10,466-10,484. DOI: 10.1002/2015JA021697

Ebihara, Y., *et al*. (2017) Possible Cause of Extremely Bright Aurora Witnessed in East Asia on 17 September 1770, *Space Weather*, **15**, 1373-1382. DOI: 10.1002/2017SW001693

Garcia, H. A., Dryer, M. (1987) The solar flares of February 1986 and the ensuing intense geomagnetic storm, *Solar Physics*, **109**, 119-137. DOI: 10.1007/BF00167403

Gonzalez, W. D., *et al*. (1994) What is a geomagnetic storm? *Journal of Geophysical Research*, **99**, A4, 5771-5792. DOI: 10.1029/93JA02867

Hapgood, H. A. (2019) The Great Storm of May 1921: An Exemplar of a Dangerous Space Weather Event, *Space Weather*, **17**, 950-975. DOI: 10.1029/2019SW002195

Hattori, K., *et al*. (2019) Occurrence of great magnetic storms on 6-8 March 1582, *Monthly Notices of the Royal Astronomical Society*, **487**, 3550-3559. DOI: 10.1093/mnras/stz1401

Hayakawa, H., *et al*. (2016) Earliest datable records of aurora-like phenomena in the astronomical diaries from Babylonia, Earth, Planets and Space, **68**, 195. DOI: 10.1186/s40623-016-0571-5

Hayakawa, H., *et al*. (2017) Records of sunspots and aurora candidates in the Chinese official histories of the Yuán and Míng dynasties during 1261-1644, *Publications of the Astronomical Society of Japan*, **69**, 65. DOI: 10.1093/pasj/psx045

Hayakawa, H., *et al*. (2018) The Great Space Weather Event during 1872 February Recorded in East Asia, *The Astrophysical Journal*, **862**, 15. DOI: 10.3847/1538-4357/aaca40

Hayakawa, H., *et al*. (2019a) Temporal and Spatial Evolutions of a Large Sunspot Group and Great Auroral Storms Around the Carrington Event in 1859, *Space Weather*, **17**, 1553-1569. DOI: 10.1029/2019SW002269

Hayakawa, H., *et al*. (2019b) The Earliest Candidates of Auroral Observations in Assyrian Astrological Reports: Insights on Solar Activity around 660 BCE, *The Astrophysical Journal Letters*, **884**, L18. DOI: 10.3847/2041-8213/ab42e4

Hayakawa, H., *et al*. (2020) The Extreme Space Weather Event in 1903 October/November: An Outburst from the Quiet Sun, *The Astrophysical Journal Letters*, DOI: 10.3847/2041-8213/ab6a18

Kawamura, A. D., *et al*. (2016) Aurora candidates from the chronicle of Qíng dynasty in several degrees of relevance, Publications of the Astronomical Society of Japan, **68**, 79. DOI: 10.1093/pasj/psw074

Lefèvre, L. (2016) Detailed Analysis of Solar Data Related to Historical Extreme Geomagnetic Storms: 1868 − 2010, *Solar Physics*, **291**, 1483-1531. DOI: 10.1007/s11207-016-0892-3

Lockwood, M., Barnard, L. (2015) An arch in the UK: a new catalogue of auroral observations made in the British Isles and Ireland, *Astron. and Geophys.*, **56**, 4.25‑4.30, DOI: 10.1093/astrogeo/atv132

Lockwood, M., et al. (2016) Tests of Sunspot Number Sequences: 2. Using Geomagnetic and Auroral Data, *Solar Physics*, **291**, 2811-2828. DOI: 10.1007/s11207-016-0913-2

Love, J. J., *et al*. (2018) Geoelectric Hazard Maps for the Mid-Atlantic United States: 100 Year Extreme Values and the 1989 Magnetic Storm, *Geophysical Research Letters*, **45**, 5-14. DOI: 10.1002/2017GL076042

Love, J. J., *et al*. (2019) Intensity and Impact of the New York Railroad Superstorm of May 1921, *Space Weather*, **17**, 1281-1292. DOI: 10.1029/2019SW002250

Nakamura, S., *et al*. (2018) Time Domain Simulation of Geomagnetically Induced Current (GIC) Flowing in 500-kV Power Grid in Japan Including a Three-Dimensional Ground Inhomogeneity, *Space Weather*, **16**, 1946-1959. DOI: 10.1029/2018SW002004

Ngwira, C. M., Pulkkinen, A. (2018) An Overview of Science Challenges Pertaining to Our Understanding of Extreme Geomagnetically Induced Currents, in: *Extreme Events in Geospace* (N. Buzulukova, ed.), 187-208. DOI: 10.1016/B978-0-12-812700-1.00008-X

Oliveira, D. M., Zesta, E. (2019) Satellite Orbital Drag During Magnetic Storms, *Space Weather*, **17**, 1510-1533. DOI: 10.1029/2019SW002287

Oliveira, D. M., *et al*. (2018) Geomagnetically Induced Currents Caused by Interplanetary Shocks With Different Impact Angles and Speeds, *Space Weather*, **16**, 636-647. DOI: 10.1029/2018SW001880

Pulkkinen, A., *et al*. (2017) Geomagnetically induced currents: Science, engineering, and applications readiness, *Space Weather*, **15**, 828-856. DOI: 10.1002/2016SW001501

Riley, P., *et al*. (2018) Extreme Space Weather Events: From Cradle to Grave, *Space Science Reviews*, **214**, 21. DOI: 10.1007/s11214-017-0456-3

Silverman, S. M., Tuan, T. F. (1973) Auroral Audibility, *Advances in Geophysics*, **16**. Edited by H. E. Landsberg and J. van Mieghem. Published by Academic Press, Inc., New York, USA, 1973, p.156

Silverman, S. M. (2008) Low-latitude auroras: The great aurora of 4 February 1872, *Journal of Atmospheric and Solar-Terrestrial Physics*, **70**, 1301-1308. DOI: 10.1016/j.jastp.2008.03.012

Silverman, S. M., Cliver, E. W. (2001) Low-latitude auroras: the magnetic storm of 14-15 May 1921, *Journal of Atmospheric and Solar-Terrestrial Physics*, **63**, 523-535. DOI: 10.1016/S1364-6826(00)00174-7

Stephenson, F. R., *et al*. (2019) Do the Chinese Astronomical Records Dated AD 776 January 12/13 Describe an Auroral Display or a Lunar Halo? A Critical Re-examination, *Solar Physics*, **294**, 36. DOI: 10.1007/s11207-019-1425-7

Störmer, C. (1955) *The Polar Aurora*, Oxford, Oxford University Press.

Tsurutani, B. T., *et al*. (2003) The extreme magnetic storm of 1-2 September 1859, Journal of Geophysical Research Space Physics, **108**, A7, 1268, DOI:

10.1029/2002JA009504

Usoskin, I. G., *et al*. (2017) An Optical Atmospheric Phenomenon Observed in 1670 over the City of Astrakhan Was Not a Mid-Latitude Aurora, *Solar Physics*, **292**, 15. DOI: 10.1007/s11207-016-1035-6

Vázquez, M., et al. (2016) Long-Term Trends and Gleissberg Cycles in Aurora Borealis Records (1600 - 2015), *Solar Physics*, **291**, 613-642. DOI: 10.1007/s11207-016-0849-6

---

## Author Comment (AC1) · 9 Apr 2020

Dear Editor, The authors want to thank the reviewers for their work on the submitted paper. In the following, specific answers to the comments are reported and all changes in the new revised manuscript are highlighted in yellow.

Reviewer #1: R1: I have carefully read the paper titled "An Early Low Latitude Aurora Observed by Rozier (Beziers, 1780)". The authors present a suspected aurora observed by Francois Rozier on 15 August 1780 in Beauséjour, close to Beziers (at MLAT= 50.18 N, according to the authors). It should be noted that the observation was made under adverse weather conditions (presence of a lightning storm). In section 4, the authors indicates that at the same time an aurora was also observed at Ratisbon

(Germany, 49 N), 5.5 further north than Beziers, and recorded in Angot's catalogue (Angot, 1897). If this article is selected for publication, I suggest some revisions to the manuscript and other small suggestions before it can be published in ANGEO.

A: Many thanks for your detailed review. We have taken into account all your suggestions and the manuscript has improved a lot.

R1: 1 Background and Introduction Line 20: For the physical mechanism of the aurora origin, (Vazquez et al. 2014) is not the appropriate reference (see e.g., Brekke, 2013, Physics of the Upper Polar Atmosphere, 2nd edn. Springer, Heidelberg).

A: We agree with the referee, We have deleted the cite of Vazquez et al. 2014 and included two more appropriate cites i.e. Brekke 2013 and Gonzalez et al., 1994.

Brekke, A.: Physics of the Upper Polar Atmosphere, 2nd Ed., (Springer) 2013.

Gonzalez, W. D., Joselyn, J. A, Kamide, Y., Kroehl, H. W., Rosoker, G., Tsuruani ,B. T. and Vasyliuna, V. M.: What is a geomagnetic storm?, J. Geophys. Res., 99, 5771-5792, doi.org/10.1029/93JA02867, 1994.

R1: Lines 25-26: The three articles cited relate to the Carrington event. It is interesting to point out other exceptional events, such as that of 1921 (Silverman, S.M., Cliver, E.W.: 2001, J. Atmos. Solar-Terr. Phys. 63, 523), as well as that which occurred in 1770 (Hayakawa, H., et al.: 2017, Astrophys. J. Lett. 850, L31).

A: We have included the events proposed by the referee and other important and well-studied events. Moreover we have updated some references of the Carrigton storm in accordance with referee 2. "This was the case of well studied extreme space weather events as those occurred on September 1770 (Hayakawa et al. 2017a); the Carrington event in August/September 1859 (Green and Boardsen, 2006; Green et al., 2006; Humble 2006; Tsurutani et al., 2003; Cliver and Dietrich, 2013; Hayakawa et al., 2019a); the storm on 1872 February (Hayakawa et al. 2018; Silverman, 2008); the extreme event on September 1909 (Hayakawa et al., 2019b); May 1921 (Hapgood,

2019; Silverman and Cliver, 2001; Love et al., 2019) or March 1989 (Allen et al., 1989) resulting in extreme magnetic disturbances and auroral displays at very low latitudes .".

Hayakawa H., Tamazawa H., Ebihara Y., Miyahara H., Kawamura A. D., Aoyama T. and Isobe H.: Records of sunspots and aurora candidates in the Chinese official histories of the Yuán and Míng dynasties during 1261–1644, Publ. Astron. Soc. Jpn 69, 65, doi: 10.1093/pasj/psx045, 2017a.

Green, J. L. and Boardsen, S.A.: Duration and extent of the great auroral storm of 1859, Adv. Space Res. 38, 130–135, 10.1016/j.asr.2005.08.054, 2006.

Green, J.L., Boardsen, S. A, Odenwald, S., Humble, J. and Pazamickas, K.A.: Eyewitness reports of the great auroral storm of 1859, Adv.Space Res. 38-2, 145-154, doi.org/10.1016/j.asr.2005.12.021, 2006.

Humble, J.: The solar events of August/September 1859 – Surviving Australian observations, Adv. Space Res. 38, 155–158, 10.1016/j.asr.2005.08.053, 2006.

Tsurutani B. T., Gonzalez W. D., Lakhina G. S., Alex S. (2003) The extreme magnetic storm of 1–2 September 1859, J. Geophys. Res., 108, 1268, doi:10.1029/2002JA009504.

Cliver, E. W. and Dietrich, W. F.: The 1859 space weather event revisited: limits of extreme activity, J. Space Weather Space Clim. 3, A31, doi: 10.1051/swsc/2013053, 2013.

Hayakawa, H., Ebihara, Y., Willis, D.M., Toriumi, S., Iju, T., Hattori, K., Wild, M. N., Oliveira, D. M., Ermolli, I., Ribeiro, J. R., Correia, A.P., Ribeiro, A. I. and Knipp, D. J: Temporal and Spatial Evolutions of a Large Sunspot Group and Great Auroral Storms Around the Carrington Event in 1859, Avd. Space Res, 17, 1553–1569. https://doi.org/10.1029/ 2019SW002269, 2019a.

Hayakawa, H., Ebihara, Y., Cliver, E. W., Hattori, K., Toriumi, S., Love, J. J., Umemura, N., Namekata, K., Sakaue, T., Takahashi, T., and Shibata, K.: The extreme space

weather event in September 1909. Monthly Notices of the Royal Astronomical Society, 484, 3, 4083-4099. DOI: 10.1093/mnras/sty3196, 2019b.

Hapgood, M.: The Great Storm of May 1921: An Exemplar of a Dangerous Space Weather Event, Adv. Space Res., 17, 950–975. https://doi.org/10.1029/2019SW002195, 2019

Love, J. J., Hayakawa, H. and Clive, E. W.: Intensity and Impact of the New York Railroad Superstorm of May 1921, Avd. Space Res, 17, 1281–1292. doi.org/10.1029/2019SW002250, 2019.

Silverman, S.M. and Cliver, E.W.: Low-latitude auroras: the magnetic storm of 14 –15 May 1921, J. Atmos. Sol-Terr. Phys. 63, 523–535, doi.org/10.1016/S1364-6826(00)00174-7, 2001

Allen, J., Frank, L., Sauer, H. and Reiff, P.: Effects of the March 1989 Solar Activity, EOS, 70, 1479-1488. doi: 10.1029/89EO004090, 1989.

R1 Line 26: About LLA, the authors state that "and have been considered a proxy of solar activity". This needs to be correctly documented. Overall, I think that this section needs to be improved and expanded with more background information.

A: We have rewritten and expanded the Background and Introduction section including an important amount of references to clarify some aspects. About the use of aurora night as a proxy we have included: "Low and mid latitude auroras nights show an association with solar activity indices as sunspot records. This link has been observed during the telescopic era (Silverman, 1992; Lockwood and Barnard, 2015; Lockwood et al., 2016) but also in pre-telescopic era from the comparison with naked-eye sunspot reports (Hayakawa et al. 2017a; Bekli and Chadou, 2019). This relationship is due mainly to the highest frequency of LMLAs during the maximum and the decaying phase of the solar cycle (Gonzalez et al., 1994). Therefore, the mid-latitude aurorae, being footprints of solar CMEs, can be considered as proxies for the long-term solar activity.

Nevertheless, LMLAs sometimes occurred in periods of low solar activity (Silverman 2003; Willis et al. 2007; Vaquero et al., 2007 and 2013; Garcia and Dryer, 1987 and Hayakawa et al., 2020). These auroras are called "sporadic auroras"".

2 Methodology 2.1. The Observer

R1: The subsection (2.1), which is a biography of F. Rozier, is unnecessary and the text should be reduced considerably.

A: The text has been shortened a 25%.

R1: Lines 35-39: please refer to reliable sources for accurate information and remove the links.

A: A new reference: Gutton, J.P. and Bonnet, J. C., Guton J. P. (Ed): Les Lyonnaises dans l'Histoire, Privat, 1991 has been added and the links eliminated as required.

2.2. The Documentary Source and the Observation description

R1 : Lines 65-66: The book's title should be rectified as follow: Observations sur la physique, sur l'histoire naturelle et sur les arts, avec des planches en taille-douce

A: The title has been amended ad indicated.

R1 : Lines 66-67: The subtitle should be rectified as follow: Observation sur une Nuée rendue phosphorique par une surabondance de l'électricité, vue de Beauséjour près de Beziers, le 15 Août

A: The subtitle has been modified as required.

R1: Lines 82-83: bad translation: The sentence "avant-coureurs de l'orage" means "before the storm" not "before it was orange colored"

A: The translation has been amended

R1: Line 93: The sentence "l'orage s'éloigna de Beziers" means "the storm moved away from Beziers", not "the orange moved away from Beziers"

A: The translation has been amended

R1: Line 101: why the author uses the term "explosion"?

A: The literary translation for Âń il n'y eut point d'explosion Âż is Âń there was no explosion". It can be also interpreted as "there was no thunder" but in those cases Rozier utilized other terms as "tonnerre".

R1: Page 2: Footnote 1: The reference must be written correctly as indicated in "Manuscript preparation guidelines for authors" of ANGEO (Publisher, Location: : :). Also, please indicate the relevant pages.

A: The footnote has been removed and the reference has been added to the reference list.

3 Analysis

R1: of the Observation Line 105: Please specify how you obtained the two values of solar depression angle (13 and 14.9).

A: The paragraph has been updated and the calculation carefully checked with the HORIZONS NASA web interface that has been quoted in the text as follows: The calculation of the solar depression angle for the geographical coordinates in Béziers and the day of the observation has been performed using the HORIZONS Web-interface of the American National Aeronautics and Space Administration (NASA) (https://ssd.jpl.nasa.gov/horizons.cgi?s_type=1#top).

R1: Lines 113-121: Color: as I said before, the orange color is not specified by the author. Therefore, this paragraph must be corrected.

A: The quote to the orange color has been cancelled and the section has been modified accordingly.

R1: Lines 133-134: There is no exact definition of the low latitude, but for me the present event must be classified as a mid-latitude aurora!

A: There is no exact definition where that boundary lies, but we agree with the referee about it is more accurate to consider the Rozier aurora as mid latitude. We have changing the text accordingly.

4 Discussion

R1: Lines 138 and 140: (Angot, 1897) not (Angot, 1896)

A: The reference has been amended: Angot A.: The Aurora Borealis, (D. Appleton & Co) 326 pp, 1897.

R1: Lines 157-160: A similar phase opposition and anti-correlation between auroral occurrence and sunspot were reported by some authors. It is an important point which must be well documented (see e.g., Silverman, S.M., 1992, Secular variation of the aurora for the past 500 Years, Rev. Geophys. 30, 333–351).

A: We have included this anticorrelation between auroral night and sunspot in the background section: "Low and mid latitude auroras nights show an association with solar activity indices as sunspot records. This link has been observed during the telescopic era (Silverman, 1992; Lockwood and Barnard, 2015; Lockwood et al., 2016) but also in pre-telescopic era from the comparison with naked-eye sunspot reports (Hayakawa et al. 2017a; Bekli and Chadou, 2019). This relationship is due mainly to the highest frequency of LMLAs during the maximum and the decaying phase of the solar cycle (Gonzalez et al., 1994). Therefore, the mid-latitude aurorae, being footprints of solar CMEs, can be considered as proxies for the long-term solar activity. Nevertheless, LMLAs sometimes occurred in periods of low solar activity (Silverman 2003; Willis et al. 2007; Vaquero et al., 2007 and 2013; Garcia and Dryer, 1987 and Hayakawa et al., 2020). These auroras are called "sporadic auroras"".

Moreover we have modify this paragraph also the paragraph commented by the referee:

"Figure 3 shows the sunspot number during the period 1766-1800. Rozier′s observation was in the declining phase of the solar cycle 3, 2-years after the maximum. This is a good moment to see LMAA because long-lived coronal holes - source of high ionized particles in the solar wind - occur more frequently in the declining phase of the sunspot cycle (Verbanac et al., 2011; Lefèvre et al., 2016). It is important to note that the Rozier's observation occurred in a period with few sunspot records. As we can see in Figure 4 the solar observations during the 1780's are rare, frequently below the 30 observations per year. For this reason, any contribution to the knowledge of the geomagnetic activity in this period is very beneficial".

R1: Overall, a more extended state of the art is needed. Some articles relating to the present work should be viewed and cited (e.g., Ordaz, J., 2010, Auroras boreales observadas en la Península Ibérica, Baleares y Canarias durante el siglo XVIII, Treb. Mus. Geol. Barcelona 17, 45-110; Legrand, J. P., & Simon, P. A., 1987, Two hundred years of auroral activity (1780-1979), AnGeo 5, 161-167; : : :)

A: The state of the art has been improved reorganizing and expanding the Background and introduction section. Both references have been included in the new version of the section.

Conclusions

R1: I think the conclusion is too short and it does not summarize the work in sufficient detail.

A: The conclusions have been rewritten as follows: "We have found a record of an atmospheric phenomenon observed on 15 August 1780 in Beausejour, close to Béziers (43° 19′ N, 3° 13′ E), France, by the abbot Francois Rozier described as a "big white cloud . . . whitish color of phosphorus burning in the open air". Rozier was not an astronomer and it is clear that he did not fully understand the phenomenon he was recording. Probably for this reason he recorded the event with minute details to later discuss it with other academicians of his time. Thanks to this accuracy, we have been able to analyse quantitative information and facts that contribute to confirm that

[Figure]

Francois Rozier observed a Mid latitude aurora that night. The aurora was observed during the nautical and astronomical twilight, it was white, enough brilliant to not be overshined by the full moon which however was above the horizon in ESE direction. It showed two bands and some rays which could fit with the class of auroral forms of both homogeneous arcs/uniform diffuse surface, and homogenous bands. Its temporal evolution could also resemble an auroral sub-storm expansion. This auroral event contributes to enlarge the geomagnetic knowledge of the late 18th century period in which the geomagnetic and the solar activity have high uncertainties due to few sunspots and LMLA observations reported from primary sources. The Rozier record is a clear case of how, a scientist from a research field far from Astronomy or Meteorology in the 18th century, could record and publish descriptions on atmospheric phenomena that he did not fully understand but however he considered worth to be documented. These sources are very valuable because they report details of infrequent and/or partially unknown atmospheric phenomena. In this case the Rozier's report had contributes to enlarge the geomagnetic knowledge of a period with low information. "

References

R1: Line 237: The source of the data (WDC-SILSO) must be cited properly as indicated on their website. I think: SILSO data/image, Royal Observatory of Belgium, Brussels. In addition, you can also indicate the version.

A: The reference has been amended.

R1: Figure 1 is not cited in the text. Furthermore, Figure 1 (b) hides part of Figure 1 (a); I think it is better to remove Figure 1 (b).

A: Figure 1 has been modified following the suggestions

Figure 1: (a) Photographic portrait of Abbot Francois Rozier (photo in public domain) (Library of Congress Prints and Photographs Division Washington; http://loc.gov/pictures/resource/ppmsca.02227/). (b) The two printed pages reporting

the aurora observation made by Abbot Francois Rozier, on 15 August 1780 in Beziers, France (Rozier, 1781).

Please also note the supplement to this comment: https://www.ann-geophys-discuss.net/angeo-2020-1/angeo-2020-1-AC1-supplement.pdf

————————————————————

Fig. 1.

---

## Author Comment (AC2) · 9 Apr 2020

Dear Editor, The authors want to thank the reviewers for their work on the submitted paper. In the following, specific answers to the comments are reported and all changes in the new revised manuscript are highlighted in yellow.

Reviewer #2:

Referee Report on MS angeo-2019-97 "An Early Low Latitude Aurora Observed by Rozier (Beziers, 1780)" by Bertolin et al.

General Comments

R2: This is an interesting case report for the mid-latitude aurora on 1780 August 15

likely at Beauséjour near Béziers. The mid-latitude auroral report in 1780 is especially important, as this will be a footprint of solar eruption and a hint for contemporary solar activity where we do not have enough coverage of sunspot observations. Overall, I think this manuscript will be an interesting contribution, while further clarifications and explanations are needed both on its background and discussions. I have listed my comments below and wish the authors to address them. Their language needs to be improved as well, preferably with professional grammatical proofreading.

A: Thank you very much for this constructive report. We answered all the comments pointed out by you. We put special attention in the reorganization and improvement of the background and the discussion. The full text of the paper was revised by an English mother tongue reviewer.

Specific Comments

1. Introduction

R2: The scientific background of this article should be improved. Rather than associating "Incursions of high-energy particles from space, mainly solar wind, strongly interact with the Earth's magnetosphere" with the cause of auroral display, I would explicitly mention the coronal mass ejection with southward interplanetary magnetic field as a cause of auroral displays in low to mid magnetic latitude.

A: We have rewritten and enlarge the background section including specific references to coronal mass ejection as cause of low to mid aurora (LMLA) as follows: "Low and Mid latitude auroras (LMLAs) are usually associated with intense space weather events, frequently caused by coronal mass ejections (CME) (Gonzalez et al., 1994; Vázquez et al. 2006). This was the case of well studied extreme space weather events as those occurred on September 1770 (Hayakawa et al. 2017a); the Carrington event in August/September 1859 (Green and Boardsen, 2006; Green et al., 2006; Humble 2006; Tsurutani et al., 2003; Cliver and Dietrich, 2013; Hayakawa et al., 2019a); the storm on 1872 February (Hayakawa et al. 2018; Silverman, 2008); the extreme event

on September 1909 (Hayakawa et al., 2019b); May 1921 (Hapgood, 2019; Silverman and Cliver, 2001; Love et al., 2019) or March 1989 (Allen et al., 1989) resulting in extreme magnetic disturbances and auroral displays at very low latitudes."

R2: I advice the authors to cite Gonzalez et al. (1994) and Daglis et al. (1999) for its references, rather than Vazquez et al. (2014).

A: References have been substituted as indicated.

Gonzalez, W. D., Joselyn, J. A, Kamide, Y., Kroehl, H. W., Rosoker, G., Tsuruani ,B. T. and Vasyliuna, V. M.: What is a geomagnetic storm?, J. Geophys. Res., 99, 5771-5792, doi.org/10.1029/93JA02867, 1994.

Daglis, I. A.: The terrestrial ring current: Origin, formation, and decay, Reviews of Geophysics, 37, 407-438, doi 10.1029/1999RG900009, 1999

R2: The space weather hazards are not only geomagnetically induced currents, but also satellite drags (Oliveira and Zesta, 2019) or atmospheric radiations (Dyer et al., 2018). These details should be cited with actual cases reports/predictions (Nakamura et al., 2018; Love et al., 2018; Boteler, 2019) and reviews (Pulkkinen et al., 2017; Ngwira and Pulkkinen, 2018; Riley et al., 2018; Oliveira et al., 2018).

A: Following the suggestion of the referee we have include some actual cases and mentioned explicitly some space weather hazards:

"It is important to note that extreme space weather events of these magnitude can provoke important impacts on our highly technological dependent society, especially in activities related with the aviation, the GPS signals, radio communication, and the electric power grid (Baker et al., 2008; Ridley et al., 2018). "

R2: Given the magnetic latitude of Béziers (50.2° MLAT), I would consider this aurora not as "low-latitude aurora" but as "mid-latitude aurora". In comparison with auroral ovals during the extreme storms like the Carrington event (∼30° MLAT), I consider that this extent is confined in mid magnetic latitude.

A: Thank you, for the comment we have considered the Rozier aurora as mid latitude, modifying the title and the text accordingly.

R2: The references on the Carrington event should be updated.

A: Thank you, the event of the 1859 described as: "the Carrington event in August/September 1859 (Green and Boardsen, 2006; Green et al., 2006; Humble 2006; Tsurutani et al., 2003; Cliver and Dietrich, 2013; Hayakawa et al., 2019a); " has the following references:

Green, J. L. and Boardsen, S.A.: Duration and extent of the great auroral storm of 1859, Adv. Space Res. 38, 130–135, 10.1016/j.asr.2005.08.054, 2006.

Green, J.L., Boardsen, S. A, Odenwald, S., Humble, J. and Pazamickas, K.A.: Eyewitness reports of the great auroral storm of 1859, Adv.Space Res. 38-2, 145-154, doi.org/10.1016/j.asr.2005.12.021, 2006.

Humble, J.: The solar events of August/September 1859 – Surviving Australian observations, Adv. Space Res. 38, 155–158, 10.1016/j.asr.2005.08.053, 2006.

Tsurutani, B. T., Gonzalez, W. D., Lakhina, G. S. and Alex, S.: The extreme magnetic storm of 1–2 September 1859, J. Geophys. Res., 108, 1268, doi:10.1029/2002JA009504, 2003.

Cliver, E. W. and Dietrich, W. F.: The 1859 space weather event revisited: limits of extreme activity, J. Space Weather Space Clim. 3, A31, doi: 10.1051/swsc/2013053, 2013.

Hayakawa, H., Ebihara, Y., Willis, D.M., Toriumi, S., Iju, T., Hattori, K., Wild, M. N., Oliveira, D. M., Ermolli, I., Ribeiro, J. R., Correia, A.P., Ribeiro, A. I. and Knipp, D. J: Temporal and Spatial Evolutions of a Large Sunspot Group and Great Auroral Storms Around the Carrington Event in 1859, Avd. Space Res, 17, 1553–1569. https://doi.org/10.1029/ 2019SW002269, 2019a.

R2: The reference of Green et al. (2014) is probably Green et al. (2006), as long as checking the NASA ADS database.

A: The referee is right, the reference has been amended.

R2: Three benchmark articles for this event should be cited in this context (Tsurutani et al., 2003; Cliver and Dietrich, 2013; Hayakawa et al., 2019a).

A: References have been added as indicated.

R2: Moreover, recent studies have located at least three rivaling storms with extremely low-latitude auroral visibility: 1872 Feb (Silverman, 2008; Hayakawa et al., 2018) and 1921 May (Silverman and Cliver, 2001; Hapgood, 2019; Love et al., 2019). These cases should be documented as well.

A: Thank you. The section has been updated as follows: "the storm on 1872 February (Hayakawa et al. 2018; Silverman, 2008); the extreme event on September 1909 (Hayakawa et al., 2019b); May 1921 (Hapgood, 2019; Silverman and Cliver, 2001; Love et al., 2019) or March 1989 (Allen et al., 1989) resulting in extreme magnetic disturbances and auroral displays at very low latitudes." Adding the quoted references:

Hayakawa, H., Ebihara, Y., Willis, D. M., Hattori, K., Giunta, A. S., Wild, M. N., Hayakawa, S., Toriumi, S.: The Great Space Weather Event during 1872 February Recorded in East Asia, The Astrophysical Journal, 862, 15. doi: 10.3847/1538-4357/aaca40, 2018.

Hayakawa, H., Ebihara, Y., Cliver, E. W., Hattori, K., Toriumi, S., Love, J. J., Umemura, N., Namekata, K., Sakaue, T., Takahashi, T., and Shibata, K.: The extreme space weather event in September 1909. Monthly Notices of the Royal Astronomical Society, 484, 3, 4083-4099. DOI: 10.1093/mnras/sty3196, 2019b.

Hapgood, M.: The Great Storm of May 1921: An Exemplar of a Dangerous Space Weather Event, Adv. Space Res., 17, 950–975. https://doi.org/10.1029/2019SW002195, 2019

Love, J. J., Hayakawa, H. and Clive, E. W.: Intensity and Impact of the New York Railroad Superstorm of May 1921, Avd. Space Res, 17, 1281–1292. doi.org/10.1029/2019SW002250, 2019.

Silverman, S.M. and Cliver, E.W.: Low-latitude auroras: the magnetic storm of 14 –15 May 1921, J. Atmos. Sol-Terr. Phys. 63, 523–535, doi.org/10.1016/S1364-6826(00)00174-7, 2001

Silverman, S.M.: Low-latitude auroras: The great aurora of 4 February 1872, J. Atmos. Sol.-Terr. Phys. 70, 1301- 1308, doi.org/10.1016/j.jastp.2008.03.012, 2008.

Allen, J., Frank, L., Sauer, H. and Reiff, P.: Effects of the March 1989 Solar Activity, EOS, 70, 1479-1488. doi: 10.1029/89EO004090, 1989.

R2: The usage of LLA as historical solar activity is another story. I would rephrase this as "Being footprints of solar eruptions, the mid-latitude aurorae (or low-latitude aurorae) are considered as proxies for the long-term solar variability", citing several relevant articles such as Silverman (1992), Lockwood and Barnard (2015), Lockwood et al. (2016), Vázquez et al. (2016), and Hayakawa et al. (2017).

A: The sentence has been re-phrased as follows: "Low and mid latitude auroras nights show an association with solar activity indices as sunspot records. This link has been observed during the telescopic era (Silverman, 1992; Lockwood and Barnard, 2015; Lockwood et al., 2016) but also in pre-telescopic era from the comparison with naked-eye sunspot reports (Hayakawa et al. 2017a; Bekli and Chadou, 2019). This relationship is due mainly to the highest frequency of LMLAs during the maximum and the decaying phase of the solar cycle (Gonzalez et al., 1994). Therefore, the mid-latitude aurorae, being footprints of solar CMEs, can be considered as proxies for the long-term solar activity. Nevertheless, LMLAs sometimes occurred in periods of low solar activity (Silverman 2003; Willis et al. 2007; Vaquero et al., 2007 and 2013; Garcia and Dryer, 1987 and Hayakawa et al., 2020). These auroras are called "sporadic auroras". "

and references have been added:

Silverman, S.M.: Secular variation of the aurora for the past 500 years, Rev. Geophys., 30, 333-351, doi.org/10.1029/92RG01571, 1992.

Lockwood, M., and Barnard, L.: An arch in the UK, Astronomy & Geophysics, 56, 4.25–4.30, doi.org/10.1093/astrogeo/atv132, 2015.

Lockwood, M., Owens, M.J., Barnard, L., Scot,t C.J., Usoskin, I.G. and Nevanlinna, H.: Tests of Sunspot Number Sequences: 2. Using Geomagnetic and Auroral Data, Sol. Phys., 291, 2811–2828, doi 10.1007/s11207-016-0913-2, 2016.

Hayakawa H., Tamazawa H., Ebihara Y., Miyahara H., Kawamura A. D., Aoyama T. and Isobe H.: Records of sunspots and aurora candidates in the Chinese official histories of the Yuán and Míng dynasties during 1261–1644, Publ. Astron. Soc. Jpn 69, 65, doi: 10.1093/pasj/psx045, 2017a.

Bekli ,M.R., and Chadou, I.: Analysis of pre-telescopic sunspots and auroras from 8th to 16th century. Adv. Space Res. 64, 1011-1018, 2019.

Gonzalez, W. D., Joselyn, J. A, Kamide, Y., Kroehl, H. W., Rosoker, G., Tsuruani ,B. T. and Vasyliuna, V. M.: What is a geomagnetic storm?, J. Geophys. Res., 99, 5771-5792, doi.org/10.1029/93JA02867, 1994.

Willis, D.M., Stephenson, F.R. and Fang H.: Sporadic aurorae observed in East Asia. Ann. Geophys., 25, 417-436, doi.org/10.5194/angeo-25-417-2007, 2007.

Vaquero, J.M., Trigo, R.M., Gallego, M.C.: Sporadic aurora from Spain. Earth Planets Space, 59, 49-51, doi.org/10.1186/BF03352061, 2007.

Vaquero, J.M., Gallego, M.C.and Domínguez-Castro, F.: A possible case of Sporadic Aurora in 1843 from Mexico. Geofísica Internacional 52, 87-92, 2013.

Garcia, H. A. and Dryer, M.: The Solar Flares of February 1986 and the Ensuing Intense Geomagnetic Storm, Sol. Phys. 109, 119-137, doi.org/10.1007/BF00167403,

1987.

Hayakawa, H., Ribeiro, P., Vaquero, J. M., Gallego, M. C., Knipp, D. J., Mekhaldi, F., Bhaskar, A., Oliveira, D. M., Notsu, Y., Carrasco, V. M. S., Caccavari, A., Veenadhari, B., Mukherjee S. and Ebihara Y.: The Extreme Space Weather Event in 1903 October/November: An Outburst from the Quiet Sun, Astrophys. J. Lett., DOI: 10.3847/2041-8213/ab6a18, 2020.

R2: Caveats must be noted here, however. Even when the solar activity is low, several great magnetic storms with significant auroral displays are reported as well (Garcia and Dryer, 1987; Hayakawa et al., 2020). This caveat should be clarified too.

A: Thank you. References have been added as indicated and the section has been updated as described in our previous answer.

2. Methodology

R2: It is nice to cite Rozier's portrait and personal detail in this article. However, citing them from wikisource or other online resources is not the best scientific practice. Please specify their original references in the publications and cite them accordingly. The reference clarifications are especially important as most of P2 of this manuscript is devoted to its explanation and the readers may wish to know more about him with appropriate references.

A: The Reference: Gutton, J.P. and Bonnet, J. C., Guton J. P. (Ed): Les Lyonnaises dans l'Histoire, Privat, 1991 has been substituted.

3. Analyses of the Observations

R2: The analyses seem sound but some improvements seem advised. The description of ""a flash started from the end of the lower area...". This is a frequent structure of the aurorae (Vaquero & Vasquez, 2009)" may be flaming of auroral display (Störmer, 1955). If the description of "a main structure of two bands oriented east to west" means westward auroral motion, this sounds consistent to the westward traveling surge (Ebihara and Tanaka, 2015).

A: The section related to the Shape of the auroral event has been completely updated as follows:

"Shape: Related to the shape description, Rozier was very accurate. The main structure described by Rozier is : "it formed a zone, a phosphoric band…at a height of 3 feet....and finally it formed an angle of 60°…. above this first luminous zone a second [zone] of the same height was formed, but with 30° of extension only i.e. half of that of the lower zone. Between one and the other a void remained, the height of which matched that one of the two connected zones". This description may fit with the report of the auroral forms class without ray structure i.e. homogeneous arcs or uniform diffuse surface, and homogenous bands too (Störmer, 1955). Nevertheless the beginning of the aurora could resemble some aspect of an auroral sub-storm expansion (Ebihara et al., 2017 and Stephenson et al., 2019) which is characterized by initial brightening of aurora, followed by a bulge expanding in all directions (Akasofu, 1964; Akasofu et al., 1965) "I noticed a luminous point…this luminous point acquired slowly [over time] volume and intensity". Moreover, Rozier records some flaming during the event "in one or in the other zone I noticed irregularities, as well as on the edges of those big white clouds [i.e. general mass or bulge]. This edge was not homogeneously bright, although in the center presented uniform bright. In the time over which the zones moved Eastward…. a flash started from the end of the lower area [of the general mass or bulge]"."

R2: The whitish auroral colour are explained with "the enhancement of the 630,0 nm [OI] emission caused by soft electrons (<100 eV) precipitating from the plasmasphere" in this manuscript. However, I suspect the whitish colour may be explained to the enhancement of the 557.7 nm of Oxygen with weak brightness or its mixture with other emissions as well (e.g., Ebihara et al., 2017; Stephenson et al., 2019). Rather than citing Abott and Juhl's statistics, it would be more straightforward to cite actual observational cases of whitish aurorae (See Section 6 of Stephenson et al., 2019).

A: The section has been updated as well as the references as follows:

"Color: He carefully mentioned the color: "whitish color of phosphorus burning in the open air". As stated by Stephenson et al. 2019 at Low-Mid geomagnetic latitude, northern lights have generally higher probability to be observed if they are reddish, however in case of an auroral display without enough brightness, it tends to appear whitish to the human eye. In addition, such effect of the human eye is enhanced if the moon is also present in the sky as the eye cannot be "dark adapted". Moreover, the whitish auroral color may be explained with the enhancement of the 557.7 nm of Oxygen with weak brightness or due to the Oxygen mixture with other emissions as well (Ebihara et al., 2017; Stephenson et al., 2019). Examples of observation which confirm that LMLA are whitish in color during extreme space weather events are reported by Ebihara et al. 2017; Green and Boardsen, 2006; Hayakawa et al. 2017b and Willis et al.1996. Rozier observed a white aurora, this made the phenomena more unusual and increase the possibility of misinterpretation of the phenomenon by Rozier himself.""

R2: For the sunspot number analyses, the authors need to cite the data source "WDC SILSO" appropriately. Likewise, the authirs need to mention the WDC SILSO in the acknowledgment.

A: The reference has been corrected and quoted as required in the website i.e. SILSO, World Data Center - Sunspot Number and Long-term Solar Observations, Royal Observatory of Belgium, on-line Sunspot Number catalogue: http://www.sidc.be/SILSO/, 'year(s)-of-data'.

In the Acknowledgment we wrote: "Credits for the use of Sunspot data to the World Data Center SILSO, Royal Observatory of Belgium, Brussels."

R2: I would strongly recommend the authors to cite Clette et al. (2014) and Clette and Lefevre, 2016) for this dataset.

A: References added in the reference list and in the text as follows:

Clette, F. and Lefèvre, L.: The New Sunspot Number: Assembling All Corrections, Sol. Phys., 291, 2629-2651, doi 10.1007/s11207-016-1014-y, 2016.

Clette, F., Svalgaard, L., Vaquero, J. M. and Cliver, E. W.: Revisiting the Sunspot Number A 400-Year Perspective on the Solar Cycle, Space Sci. Rev.,186, 35–103, doi10.1007/s11214-014-0074-2, 2014.

R2: I suspect that the cause of this storm is probably better explained with the coronal mass ejections (see Gonzalez et al., 1994; Daglis et al., 1999) rather than the high-speed solar wind from the corona hole.

A: This was clarigied immediately in the backround and introduction: "Low and Mid latitude auroras (LMLAs) are usually associated with intense space weather events, frequently caused by coronal mass ejections (CME) (Gonzalez et al., 1994; Daglis, 1999; Vázquez et al. 2006). "

Conclusion

R2: The conclusion needs to be more developped to be an independent original article.

A: The conclusions have been rewritten as follows:

" We have found a record of an atmospheric phenomenon observed on 15 August 1780 in Beausejour, close to Béziers (43° 19′ N, 3° 13′ E), France, by the abbot Francois Rozier described as a "big white cloud . . . whitish color of phosphorus burning in the open air". Rozier was not an astronomer and it is clear that he did not fully understand the phenomenon he was recording. Probably for this reason he recorded the event with minute details to later discuss it with other academicians of his time. Thanks to this accuracy, we have been able to analyse quantitative information and facts that contribute to confirm that Francois Rozier observed a Mid latitude aurora that night. The aurora was observed during the nautical and astronomical twilight, it was white, enough brilliant to not be overshined by the full moon which however was above the horizon in ESE direction. It showed two bands and some rays which could fit
with the class of auroral forms of both homogeneous arcs/uniform diffuse surface, and homogenous bands. Its temporal evolution could also resemble an auroral sub-storm expansion. This auroral event contributes to enlarge the geomagnetic knowledge of the late 18th century period in which the geomagnetic and the solar activity have high uncertainties due to few sunspot and LMLA observations reported from primary sources. The Rozier record is a clear case of how, a scientist from a research field far from Astronomy or Meteorology in the 18th century, could record and publish descriptions on atmospheric phenomena that he did not fully understand but however he considered worth to be documented. These sources are very valuable because they report details of infrequent and/or partially unknown atmospheric phenomena. In this case the Rozier's report had contributes to enlarge the geomagnetic knowledge of a period with low information. "

Supplement

R2: It may be helpful to reproduce the French text here, as the authors stated "complete original French version is reported in the Supplementary Materials".

A: Both pages of the original text have been added in Figure 1. The sentence in the manuscript has been corrected consequently.

Minor Comments

R2: P1L12: The authors need to be consistent for the usage of Béziers or Bezier.

A: The text has been corrected using Béziers all times.

R2: By the ways, isn't the observational site Beauséjour? If this is the case, the coordinate should be N43°19′, E3°13′.

A: Corrected

R2: P1L20: disturbs => disturbances

A: The term has been amended as indicated.

R2: P1L22: Babylonians (Stephenson et al., 2004) => Assyrians and Babylonians (Stephenson et al., 2004; Hayakawa et al., 2016, 2019b)

A: Text has been integrated as required as well as references.

R2: P1L30: "require an accurate analysis to avoid possible misinterpretations" => Cite Kawamura et al. (2016), Usoskin et al. (2017), and Stephenson et al. (2019) here.

A: References have been added as required i.e.

Kawamura A.D., Hayakawa H., Tamazawa H., Miyahara H. and Isobe H.: Aurora candidates from the chronicle of Qíng dynasty in several degrees of relevance, Publ. Astron. Soc. Japan, 68, 79, doi: 10.1093/pasj/psw074, 2016.

Usoskin, I.G., Kovaltsov, G.A., Mishina, L.N., Sokoloff, D.D. and Vaquero, J.: An Optical Atmospheric Phenomenon Observed in 1670 over the City of Astrakhan Was Not a Mid-Latitude Aurora, Sol. Phys., 292, 15. DOI: 10.1007/s11207-016-1035-6, 2017.

Stephenson, F. R., Willis D. M., Hayakawa, H., Ebihara, Y., Scott, C. J., Wilkinson, J. Wild, M. N.: Do the Chinese Astronomical Records Dated AD 776 January 12/13 Describe an Auroral Display or a Lunar Halo? A Critical Re-examination, Sol. Phys. 294, 36, doi.org/10.1007/s11207-019-1425-7, 2019.

R2: P2: Please italicise the journal titles.

A: Journal titles have been italicized.

R2: P2L48: Béziers (Beauséjour) => Beauséjour in the suburb of Béziers

A: The sentence has been modified as indicated.

R2: P3L68: "reported below in English" => "reported below in our English translation"

A: The expression has been modified as indicated.

R2: P3L95: "proved" => "proven"

A: The error has been corrected.

R2: P3L104: "the measure of time of the French Hours that lasted until the French Revolution in 1789" => Please cite a reference for this statement.

A: the text has been modified as follows:" Hour of Observation and sun depression angle: Rozier describes the starting (20:05) and ending (20:17) hour of his observation as local solar time (LST) i.e. the measure of local time as in use in the XVIII century. The pendulum clocks locally could be synchronized following the daily data reported in the Ephemerides with the time of the sunrise, midday and sunset published yearly (Jeaurat, 1780). "

The Ephemerides from the Observatory of Paris for the year of the observation of Rozier has been quoted: "Jeaurat, E.S : Connoissance des Temps pour l'Année bis-sextile 1780, Publiée Par l'ordre de l'Académie Royale des Sciences, et calculée par M. Jeaurat, de la meme Académie. De l'Imprimerie Royale, Paris, 1780. Âż

R2: P3L104: "These times correspond" => "Given its longitude, these time stamps correspond"

A: The sentence has been integrated as indicated.

R2: P3L105: "nautical twilight" => "astronomical twilight"

A: Done.

R2: P4L122-123: For auroral audibility, cite the review of Silverman and Tuan (1973).

A: Done. The section has been rewritten as follows:" Noise: Silverman and Tuan (1973) said that from observational evidence, the most likely sound accompany auroral observations could derive from discharges generated by aurorally associated electric fields. Rozier, although in his observation reported: "It appeared [to me] that these areas were a simple mass of steam, only charged by electricity, which it made them transparent and phosphoric". However, he concluded saying that: "for three different times,

a flash, with almost null noise, started from the end of the lower area [i.e. the bulge] . . . [and again] when the light flashed . . .there was no blast". This absence of sound recorded by Rozier discard a possible misinterpretation with other noisy atmospheric phenomena." The reference has been added as required:

Silverman, S.M. and Tuan, T.F.: Auroral Audibility, Ad. Geophys., 16,155-266, doi.org/10.1016/S0065-2687(08)60352-0, 1973.

R2: P5L124-128: For bright aurorae visible during night with the full moon, it would be advised to reinforce the existing discussions with actual observational cases cited in Stephenson et al. (2019) and Hattori et al. (2019).

A: The section has been updated as follows:" Moon: whether or not an aurora is overshined by the moon depends on the lunar phase, the brightness of the aurora, and the angular distance between the moon and the sky position occupied by auroral emission (Stephenson et al. 2019). Rozier does not report any information about the moon. But the moon was in the sky that day. The moon, on 15 August 1780 was full moon and rose at 19:25 (UT) at an azimuth angle of 111.4° ESE direction, i.e. opposite respect the direction of observation of Rozier and close to the horizon. During the time Rozier observed the phenomenon, the moon was at azimuth angle 116.5° and elevation angle 3.4° (at 19:55 UT) while at the end of his observation it was at azimuth angle 118.5° and elevation angle 5.3° (at 20:07 UT) therefore always in the direction ESE. The short time of the observation suggests that although the aurora was highly bright because Rozier could record it with full moon in the sky (Stephenson et al., 2019; Hattori et al., 2019), however because of the moon rise above the horizon the light conditions could hinder the visibility of the aurora as well as the presence of tropospheric clouds. In literature several auroral observations are reported during full moon e.g. those observed on the 18 February and on 12 November 1837 (Olmsted, 1837; Snow, 1842), those reported by Martin, 1847 and Glaisher, 1847 on October 24 of that year, and the event observed on 4 September 1908 described by Barnard, 1910."

Reference added in the reference list:

Stephenson, F. R., Willis D. M., Hayakawa, H., Ebihara, Y., Scott, C. J., Wilkinson, J. Wild, M. N.: Do the Chinese Astronomical Records Dated AD 776 January 12/13 Describe an Auroral Display or a Lunar Halo? A Critical Re-examination, Sol. Phys. 294, 36, doi.org/10.1007/s11207-019-1425-7, 2019.

Hattori, K., Hayakawa, H. and Ebihara, Y.: Occurrence of Great Magnetic Storms on 6–8 March 1582, Mon. Not. R. Astron. Soc., 487, 3550–3559, doi.org/10.1093/mnras/stz1401, 2019.

Olmsted, D.: Observations on the aurora borealis on Jan. 25, 1837 Am. J. Sci. Arts 32, 176, 1837.

Snow, R.: Observations of the Aurora Borealis. From September 1834 to September 1839, Moyes & Barclay, London,1842.

Barnard, E.E.: Observations of the aurora, made at the Yerkes Observatory, 1902 – 1909. Astrophys. J. 31, 208, 1910.

R2: P5L137: "15th august 1780" => "15 August 1780".

A: The date has been corrected.

R2: P5L143: "is" => "it is"

A: The subject has been added.

R2: P6L157: "decrease phase" => "declining phase"

A: The expression has been modified as indicated.

R2: P6L158: "2-years delayed respect the peak of the highest solar activity" => "2-years after the maximum"

A: The sentence has been modified as required.

R2: P6L160: Cite Lefevre et al. (2016) here.

A: The reference has been added: Lefèvre, L.: Detailed Analysis of Solar Data Related to Historical Extreme Geomagnetic Storms: 1868 – 2010, Sol. Phys., 291, 1483-1531. DOI: 10.1007/s11207-016-0892-3, 2016.

R2: P6L160: "few solar observation" => "few sunspot observations"

A: The word has been substituted as indicated.

R2: P6L162: "very welcomed" => "useful" or "informative"

A: The word has been substituted as indicated

R2: P6L164: "this solar observation" => maybe "the nearest solar observation"?

A: Done

R2: P6L166: "This is 48 days without sunspot information" => "It means this event occurred in an interval without sunspot data for 48 days".

A: The sentence has been modified as indicated.

R2: Figure 1: The two figures are overlapped. They should be separated at least.

A: Both pages of the document are reported in figure 1 instead of Fig. 1a and Fig 1b.

R2: The data source of the photograph should be addressed not with the URL but with shelf mark in the Library of Congress Prints and Photographs Division Washington.

A: Done

R2 : Figure 3: Cite Clette et al. (2014) and Clette and Lefevre (2016) here.

A: References have been added:

Clette, F. and Lefèvre, L.: The New Sunspot Number: Assembling All Corrections, Sol. Phys., 291, 2629-2651, doi 10.1007/s11207-016-1014-y, 2016.

[Figure]

Clette, F., Svalgaard, L., Vaquero, J. M. and Cliver, E. W.: Revisiting the Sunspot Number A 400-Year Perspective on the Solar Cycle, Space Sci. Rev.,186, 35–103, doi10.1007/s11214-014-0074-2, 2014.

R2. Figure 4: Cite Dominguez-Castro et al. (2016) and Vaquero et al. (2016) here.

A: References have been added:

Domínguez-Castro, F., Vaquero, J.M., Bertolin, C., Gallego, M. C., De la Guia, C. and Camuffo, D. : Aurorae observed by Giuseppe Toaldo in Padua (1766-1797), J. Space Weather Spac., 6, A21, doi.org/10.1051/swsc/2016016, 2016.

Vaquero, J.M., Svalgaard, L., Carrasco, V.M.S., Clette, F., Lefèvre, L., Gallego, M.C., Arlt, R., Aparicio, A.J.P., Richard, J-G. and Howe, R.: A revised collection of sunspot group numbers. Sol. Phys. 291, 3061-3074, 10.1007/s11207-016-0982-2, 2016.

Please also note the supplement to this comment:
https://www.ann-geophys-discuss.net/angeo-2020-1/angeo-2020-1-AC2-supplement.pdf

Fig. 1.

[Figure]

---

## Referee Report (RR1)

**Second Referee Report on MS Angeo-2020-1**

**Summary**

-- This manuscript has been significantly improved and I am happy to recommend its publication. This case report forms an interesting analysis for space weather event in 1780 within the scarcity of solar observations at that time and improve our understanding of the transitions from the high solar cycles to the Dalton Minimum. Apart from my concerns on the sporadic aurorae, most of my comments are rather minor. I hope they could be of use to improve this manuscript.

**Major Comment**

-- 1. Sporadic aurorae

-- The authors' statement on the "sporadic auroras" is not true. Here, I have meant there are some major geomagnetic storms even around the cycle minima or their immediate aftermath (Garcia and Dryer, 1987; Hayakawa et al., 2020c). On the other hand, sporadic aurorae are aurorae seen in mid to low magnetic latitude during moderate to quiet geomagnetic activity, as explained in Silverman (2003). Similar reports are found globally (Willis et al., 2007; Vaquero et al., 2007, 2011; Hayakawa et al., 2018; Oliveira et al., 2020). While their physical nature is not extremely certain, part of them are explained with short storms and CIR-storms (see Hayakawa et al., 2018; Bhaskar et al., 2020). Difference of major geomagnetic storms around the solar minima and sporadic aurorae should be clarified, to avoid any potential confusions.

**Minor Comments**

-- Title: Bezier => Beausejour

-- P1L8 Aurorae Observations => Auroral observations

-- P1L12 Mid => mid

-- P1L27 The term of "Carrington event" generally describes

-- P1L29 For May 1921 storm and March 1989 storm, please cite Hayakawa et al. (2019a) and Boteler et al. (2019), respectively.

-- P2L33 Ridley => Riley

-- P2L34 auroras nights => aurorae

-- P4L126 Some bright aurorae are seen even under twilight. You could cite Silverman and Cliver (2001), Vaquero et al. (2008), and Hayakawa et al. (2019) for your explanation.

-- P5L144-146 The sentence is too long and difficult to read. The authors should separate them and keep their sentence readable.

-- P5L151-153 This sentence should be sent to the beginning of this section and continued with "However, as stated by Stephenson et al. (2019) ...".

-- P6L170 The presence of the full moon may have fainted the auroral display and made it apparently whitish due to their relative brightness. This possibility may make the authors discussions more plausible.

-- P6L183 According with => According to

-- P7L210 The scarcity of solar observations makes this study extremely important. This should be emphasised. Understanding of the space climate in this period improves our discussions on the transition from the high solar cycles to the Dalton Minimum (Usoskin et al., 2009; Karoff et al., 2015; Owens et al., 2015; Hayakawa et al., 2020b).

-- P10L296 The solar depression angle should be done with the coordinate of Beausejour, not that of Bezier, although the resultant difference should be extremely minor.

-- Figure 2: Auroras Observations => Auroral Observations

-- Figure 4: The usage of sunspot number should appropriately credit SILSO (Clette et al., 2014; Clette and Lefevre, 2016) here.

**Additional References**

Bhaskar, A., et al.: 2020, An analysis of the Trouvelot's Auroral Drawing on 1/2 March 1872: Plausible Evidence for Recurrent Geomagnetic Storms, Journal of Geophysical Research: Space Physics, 125, e2020JA028227. DOI: 10.1029/2020JA028227

Hayakawa, H., Vaquero, J. M., Ebihara, Y.: 2018, Sporadic auroras near the geomagnetic equator: in the Philippines, on 27 October 1856, Annales Geophysicae, 36, 1153-1160. DOI: 10.5194/angeo-36-1153-2018

Hayakawa, H., et al.: 2019, The Celestial Sign in the Anglo-Saxon Chronicle in the 770s: Insights on Contemporary Solar Activity, Solar Physics, 294, 42. DOI: 10.1007/s11207-019-1424-8

Hayakawa, H., et al.: 2020b, Thaddäus Derfflinger's Sunspot Observations during 1802—1824: A Primary Reference to Understand the Dalton Minimum, The Astrophysical Journal, 890, 98. DOI: 10.3847/1538-4357/ab65c9

Karoff, C., et al.: 2015, The lost sunspot cycle: New support from $^{10}$Be measurements, Astronomy & Astrophysics, 575, A77. DOI: 10.1051/0004-6361/201424927

Oliveira, D. M., et al.: 2020, A possible case of sporadic aurora observed at Rio de Janeiro, Earth,

Planets and Space, 72, 82. DOI: 10.1186/s40623-020-01208-z

Owens, M. J., et al.: 2015, The heliospheric Hale cycle over the last 300 years and its implications for a "lost" late 18th century solar cycle, Journal of Space Weather and Space Climate, 5, A30. DOI: 10.1051/swsc/2015032

Usoskin, I. G., et al.: 2009, A Solar Cycle Lost in 1793-1800: Early Sunspot Observations Resolve the Old Mystery, The Astrophysical Journal Letters, 700, L154-L157.

Vaquero, J. M., et al.: 2008, The 1870 space weather event: Geomagnetic and auroral records, Journal of Geophysical Research: Space Physics, 113, A8, A08230. DOI: 10.1029/2007JA012943

---

## Referee Report (RR2)

**Referee Report**

I have read the revised version of the manuscript submitted by Bertolin et al., retitled as "An Early Mid Latitude Aurora Observed by Rozier (Béziers, 1780)", as well as their responses to the two reviewers who assessed the initial version.

In this manuscript, the authors analyzed the report of Francois Rozier concerning the observation of an aurora borealis in Beauséjour, close to Beziers (at MLAT= 50.18°N). The event is described as a "big white cloud … whitish color of phosphorus burning in the open air". It occurred on 15 August 1780 in the declining phase of the solar cycle 3, as shown in Figure 2. In this period, no sunspots were seen on the disk of the Sun for 48 days (Figure 4). However, it should be noted that during the same night (15 Aug.), the aurora was also observed at Ratisbon (Germany, 49º 01' N, 12° 05' E) and recorded in Angot's catalogue (1897).

In section 3, the physical description of the aurora is analyzed (shape and color). The authors noted that the event occurred during the full moon phase, which has not been mentioned by Rozier.

In Section 4, a comparative analysis is made with other existing series recorded by Toaldo (Padova), Salvà (Barcelona), and Hughes (Stroud). They noticed that the Rozier's observation was close to the maximum observed by Toaldo.

In conclusion, the authors indicate the importance of this record: "*This auroral event contributes to enlarge the geomagnetic knowledge of the late 18th century period in which the geomagnetic and the solar activity have high uncertainties due to few sunspot and LMLA observations reported from primary sources*".

The authors answered correctly to the questions of the two reviewers. The text of the manuscript improved, many errors corrected, and the main approach revised. The introduction and the conclusion especially are rewritten and expanded.

The subsection (2.1), which is a biography of F. Rozier, is reduced by about quarter and a new reference is added.

**Minor Comments:**

I think that the paper can be accepted for publication in ANGEO after considering some minor comments:

1. Line 135: "homogenous bands tooStörmer (1955)", correct the sentence
2. Figure 1 (a) : that is not a "Photographic portrait", I think is an "Engraved portrait". Also, I cannot find this photo on the website of the Library of Congress (loc.gov) !!! Please verify all information is correct
3. Figure 1 (b): The source of the two printed pages reporting the aurora must be cited as "Source gallica.bnf.fr/Bibliothèque Nationale de France"
4. Legend of Figure 2: instead of "The grey column remarks the year …" I prefer "The grey column indicates the year …"
5. References to books shall be made in the forms indicated in "Manuscript preparation guidelines for authors" of ANGEO: https://www.annales-geophysicae.net/for_authors/manuscript_preparation.html
6. In Supplementary Material, the text must be correctly displayed. Furthermore, it is not necessary to cite the same reference two times.
7. I am not native English speaker, but I think the language should be improved throughout the manuscript.

---

## Author Response (AR2)

Dear Editor,

The authors want to thank the reviewers for their work on the submitted paper.

In the following, specific answers to the comments are reported and all changes in the new revised manuscript are highlighted in yellow.

**Reviewer #1:**

**Second Referee Report on MS Angeo-2020-1**

**Summary**

R1: -- This manuscript has been significantly improved and I am happy to recommend its publication. This case report forms an interesting analysis for space weather event in 1780 within the scarcity of solar observations at that time and improve our understanding of the transitions from the high solar cycles to the Dalton Minimum. Apart from my concerns on the sporadic aurorae, most of my comments are rather minor. I hope they could be of use to improve this manuscript.

*A: Many thanks for your comments of appreciations of the work done.*

**Major Comment**

R1: -- 1. Sporadic aurorae -- The authors' statement on the "sporadic auroras" is not true. Here, I have meant there are some major geomagnetic storms even around the cycle minima or their immediate aftermath (Garcia and Dryer, 1987; Hayakawa et al., 2020c). On the other hand, sporadic aurorae are aurorae seen in mid to low magnetic latitude during moderate to quiet geomagnetic activity, as explained in Silverman (2003). Similar reports are found globally (Willis et al., 2007; Vaquero et al., 2007, 2011; Hayakawa et al., 2018; Oliveira et al., 2020). While their physical nature is not extremely certain, part of them are explained with short storms and CIR-storms (see Hayakawa et al., 2018; Bhaskar et al., 2020).

Difference of major geomagnetic storms around the solar minima and sporadic aurorae should be clarified, to avoid any potential confusions.

*A: The reviewer is completely correct. To avoid misunderstandings, we have decided to delete this sentence about sporadic aurorae. The article is not focused on this kind of auroras, and any discussion about this kind of events is completely unnecessary.*

**Minor Comments**

R1:-- Title: Bezier => Beausejour

*A: Corrected.*

R1:-- P1L8 Aurorae Observations => Auroral observations

*A: Corrected, also in in the text at line 10P1.*

R1:-- P1L12 Mid => mid

*A: Corrected.*

R1:-- P1L27 The term of "Carrington event" generally describes

*A: Corrected.*

R1:-- P1L29 For May 1921 storm and March 1989 storm, please cite Hayakawa et al. (2019a) and Boteler et al. (2019), respectively.

*A: Done*

R1:-- P2L33 Ridley => Riley

*A: Corrected*

R1:-- P2L34 auroras nights => aurorae

*A: Corrected*

R1:-- P4L126 Some bright aurorae are seen even under twilight. You could cite Silverman and Cliver (2001), Vaquero et al. (2008), and Hayakawa et al. (2019) for your explanation.

*A: Corrected in lines 123-124 at page 4*

R1:-- P5L144-146 The sentence is too long and difficult to read. The authors should separate them and keep their sentence readable.

*A: We have improved the sentences as follows: In fact, Rozier reports: "I noticed a luminous point…this luminous point acquired slowly [over time] volume and intensity". Furthermore, he records some flaming during the event "in one or in the other zone I noticed irregularities, as well as on the edges of those big white clouds [i.e. general mass or bulge]. He finally adds more details: "This edge was not homogeneously bright, although in the center presented uniform bright. In the time over which the zones moved Eastward…. a flash started from the end of the lower area [of the general mass or bulge]".*
*at page 5 lines 137-141.*

R1:-- P5L151-153 This sentence should be sent to the beginning of this section and continued with "However, as stated by Stephenson et al. (2019) …".

*A: The original sentence has been reformulated in: "As stated by Stephenson et al. 2019 at Low-Mid geomagnetic latitude, northern lights have generally higher probability to be observed when they are reddish. While, when they appear whitish to the human eye, the aurorae have lower brightness." at page 5 lines 144-145.*

R1:-- P6L170 The presence of the full moon may have fainted the auroral display and made it apparently whitish due to their relative brightness. This possibility may make the authors discussions more plausible.

*A: Thanks for this feedback, we have added it: "However, because of the moon rise above the horizon, the light conditions could have fainted the auroral display and made it apparently whitish due to their relative brightness. " in the text at line: 167-168 page 6.*

R1:-- P6L183 According with => According to

*A: Corrected at line 182 page 6.*

R1:-- P7L210 The scarcity of solar observations makes this study extremely important. This should be emphasised. Understanding of the space climate in this period improves our discussions on the transition from the high solar cycles to the Dalton Minimum (Usoskin et al., 2009; Karoff et al., 2015; Owens et al., 2015; Hayakawa et al., 2020b).

*A: Thanks for this comment. This has been inserted in the text at lines 209-211, page 7.*

R1:-- P10L296 The solar depression angle should be done with the coordinate of Beausejour, not that of Bezier, although the resultant difference should be extremely minor.

*A: Corrected, the solar depression angle values calculated at the coordinate of Beausejour, and not that of Bezier have been substituted in the text at line: 122, page 4.*

R1:-- Figure 2: Auroras Observations => Auroral Observations

*A: Corrected, a new figure has been done.*

R1:-- Figure 4: The usage of sunspot number should appropriately credit SILSO (Clette et al., 2014; Clette and Lefevre, 2016) here.

*A: The suggested credit has been reported in the caption of Figure 4.*

**Additional References**

R1: Bhaskar, A., et al.: 2020, An analysis of the Trouvelot's Auroral Drawing on 1/2 March 1872: Plausible Evidence for Recurrent Geomagnetic Storms, Journal of Geophysical Research: Space Physics, 125, e2020JA028227. DOI: 10.1029/2020JA028227

Hayakawa, H., Vaquero, J. M., Ebihara, Y.: 2018, Sporadic auroras near the geomagnetic equator: in the Philippines, on 27 October 1856, Annales Geophysicae, 36, 1153-1160. DOI: 10.5194/angeo-36-1153-2018

Hayakawa, H., et al.: 2019, The Celestial Sign in the Anglo-Saxon Chronicle in the 770s: Insights on Contemporary Solar Activity, Solar Physics, 294, 42. DOI: 10.1007/s11207-019-1424-8

Hayakawa, H., et al.: 2020b, Thaddäus Derfflinger's Sunspot Observations during 1802–1824: A Primary Reference to Understand the Dalton Minimum, The Astrophysical Journal, 890, 98. DOI: 10.3847/1538-4357/ab65c9

Karoff, C., et al.: 2015, The lost sunspot cycle: New support from 10Be measurements, Astronomy & Astrophysics, 575, A77. DOI: 10.1051/0004-6361/201424927

Oliveira, D. M., et al.: 2020, A possible case of sporadic aurora observed at Rio de Janeiro, Earth,

3 Planets and Space, 72, 82. DOI: 10.1186/s40623-020-01208-z

Owens, M. J., et al.: 2015, The heliospheric Hale cycle over the last 300 years and its implications

for a "lost" late 18th century solar cycle, Journal of Space Weather and Space Climate, 5, A30. DOI:

10.1051/swsc/2015032

Usoskin, I. G., et al.: 2009, A Solar Cycle Lost in 1793-1800: Early Sunspot Observations Resolve

the Old Mystery, The Astrophysical Journal Letters, 700, L154-L157.

Vaquero, J. M., et al.: 2008, The 1870 space weather event: Geomagnetic and auroral records,

Journal of Geophysical Research: Space Physics, Dear Editor,

The authors want to thank the reviewers for their work on the submitted paper.

In the following, specific answers to the comments are reported and all changes in the new revised manuscript are highlighted in yellow.

*A:Many thanks, we have added the references quoted in your revision and checked the whole bibliography of the manuscript.*

**Reviewer #2:**

**Referee Report**

R2: I have read the revised version of the manuscript submitted by Bertolin et al., retitled as "An Early Mid Latitude Aurora Observed by Rozier (Béziers, 1780)", as well as their responses to the two reviewers who assessed the initial version.

In this manuscript, the authors analyzed the report of Francois Rozier concerning the observation of an aurora borealis in Beauséjour, close to Beziers (at MLAT= 50.18°N). The event is described as a "big white cloud … whitish color of phosphorus burning in the open air". It occurred on 15 August 1780 in the declining phase of the solar cycle 3, as shown in Figure 2. In this period, no sunspots were seen on the disk of the Sun for 48 days (Figure 4). However, it should be noted that during the same night (15 Aug.), the aurora was also observed at Ratisbon (Germany, 49º 01' N, 12° 05' E) and recorded in Angot's catalogue (1897).

In section 3, the physical description of the aurora is analyzed (shape and color). The authors noted that the event occurred during the full moon phase, which has not been mentioned by Rozier.

In Section 4, a comparative analysis is made with other existing series recorded by Toaldo (Padova), Salvà (Barcelona), and Hughes (Stroud). They noticed that the Rozier's observation was close to the maximum observed by Toaldo.

In conclusion, the authors indicate the importance of this record: "This auroral event contributes to enlarge the geomagnetic knowledge of the late 18th century period in which the geomagnetic and the solar activity have high uncertainties due to few sunspot and LMLA observations reported from primary sources".

The authors answered correctly to the questions of the two reviewers. The text of the manuscript improved, many errors corrected, and the main approach revised. The introduction and the conclusion especially are rewritten and expanded. The subsection (2.1), which is a biography of F. Rozier, is reduced by about quarter and a new reference is added.

*A:Tthanks for the appreciation of the work done.*

**Minor Comments:**

R2: I think that the paper can be accepted for publication in ANGEO after considering some minor comments:

*A: Thanks*

R2: 1. Line 135: "homogenous bands tooStörmer (1955)", correct the sentence

*A: Corrected*

2. Figure 1 (a) : that is not a "Photographic portrait", I think is an "Engraved portrait". Also, I cannot find this photo on the website of the Library of Congress (loc.gov) !!! Please verify all information is correct

*A: thanks for the clarification. We have modified Figure 1 deciding to erase the portrait of the abbot Rozier as It was difficult to find the original image we decide to keep only the original analyzed text in the new Figure 1.*

3. Figure 1 (b): The source of the two printed pages reporting the aurora must be cited as "Source gallica.bnf.fr/Bibliothèque Nationale de France"

*A: see comments above.*

4. Legend of Figure 2: instead of "The grey column remarks the year …" I prefer "The grey column indicates the year …"

*A: Thanks, done.*

5. References to books shall be made in the forms indicated in "Manuscript preparation guidelines for authors" of ANGEO:

https://www.annalesgeophysicae.net/for_authors/manuscript_preparation.html

*A: Checked and corrected.*

6. In Supplementary Material, the text must be correctly displayed. Furthermore, it is not necessary to cite the same reference two times.

*A: Thanks. We have used the same font and font size of the manuscript and cancelled the references that are already reported in the manuscript.*

7. I am not native English speaker, but I think the language should be improved throughout the manuscript.

*A: we have revised the whole manuscript to improve the English language.*

**An Early Mid Latitude Aurora Observed by Rozier (==Beausejour==, 1780)**

Chiara Bertolin[1], Fernando Domínguez-Castro[2,3], Lavinia. de Ferri[1]

[revised manuscript text omitted]

---

## Author Response (AR3)

**Minor Revision**

Topical Editor Decision: Publish subject to technical corrections (20 Sep 2020) by Margit Haberreiter

Comments to the Author:

Dear authors, I am delighted to inform you that your manuscript "An Early Mid Latitude Aurora Observed by Rozier (Beausejour, 1780)" is accepted for publication in Annales Geophysicae.

However, may I ask you to address the following editorial remarks:

R1: Section 2 Methodology

At the beginning and end of the section:

Line 55/56: there are two lines with numbers " 1. , 2. ". Please delete those.

Seme in Lines 77/78

A: done

R1: Section 2.1 is empty; number should be deleted and subsequent numbering corrected accordingly.

A: done

R1: Section 3 Analysis of the Observation

For better readibility, I recommend to put the the titles of the paragraphs (Hour of observation and sun depression angle; Shape:; etc) in bold font (not underlined).

A: done

R1: Paragraph Moon: "whether" -> "Whether"

A: done

R1: References:

- Domínguez-Castro, F., Vaquero, J.M., Bertolin, C., Gallego, M. C., De la Guia, C. and Camuffo, D. : Aurorae observed by

Giuseppe Toaldo in Padua (1766-1797), J. Space Weather Spac ., 6, A21, doi: 10.1051/290 swsc/2016016, 2016.

Please delete space after "Camuffo, D." and before ":"

A: done

R1: - HORIZONS Web-interface of the American National Aeronautics and Space Administration (NASA)

(https://ssd.jpl.nasa.gov/horizons.cgi?s_type=1#top). Visited 11th August 2020

The reference to the "HORIZONS Web-interface" would be better suited in the acknowledgement section. Also, the date when the webpage was visited ("Visited 11th August 2020") should not be mentioned. if there are subsequent data versions, the version could be mentioned.

A: done

R1: - Library of Congress Prints and Photographs Division Washington, D.C. 20540 USA DIGITAL ID: (digital file from original

print) http://loc.gov/pictures/resource/ppmsca.02227/ (accessed 06/01/ 2020) -> Please remove "accessed 06/01/ 2020)"

A: done

R1: - SILSO, World Data Center - Sunspot Number and Long-term Solar Observations, Royal Observatory of Belgium, on-line

Sunspot Number catalogue: http://www.sidc.be/SILSO/, 'year(s)-of-data' -> the link does not seem to work. Please enter correct link. The addition " 'year(s)-of-data' " should be removed.

A: corrected

R1: Line 395 is empty -> Pleae remove.

A: done

R1: General comments on the Figures: The text of the figure captions should not be in bold font. The title of the y-axes should be shortened (further details see below).

A: corrected

R1: Figure 1: The text of the figure captions should not be in bold font. The figure seems cut on the right-hand side. The reference to Google Books is not necessary, as the correct reference is already given.

A: corrected

R1: Figure 2: The title of the x-axis is rather Time (year) than "Date" (as day and month is not given)

A: corrected

R1: Figure 2: Left y-axis: Geomagnetic Latitude variations -> also give units: (deg); The title of the right y-axis "Yearly total Auroral Observations" -> "Number of Auroras"; the full meaning should be clearly given in the figure caption.

A: corrected

R1: Figure 3: Left y-title: "Yearly Total Sunspot Number"-> "Sunspot Number" (details should be given in the figure caption)

Figure 3: Right y-title: "Number of days with solar observations" -> "number of days";

A: corrected

R1: Figure 3: Two references seem a lot for the figure. If the figure is not adopted from a publication, a reference is not needed in the figure caption. The details of the data used should be explained in the text along with the appropriate references. In Figure 3 (and Figure 4) it is then sufficient to just use "SILSO-WDC sunspot number"

A: done

R1: Figure 4: The references to "(Dominguez-Castro et al., 2016; Vaquero et al. 2016)" as the data set should be/is already introduced in the main text and references here accordingly.

Generally, the text of the Figure captions does not need to be in bold font.

A: corrected